# A Generalized Neural Tangent Kernel Analysis for Two-layer Neural Networks

**Zixiang Chen**
Department of Computer Science
University of California, Los Angeles
Los Angeles, CA 90095, USA
chenzx19@cs.ucla.edu

**Yuan Cao**
Department of Computer Science
University of California, Los Angeles
Los Angeles, CA 90095, USA
yuancao@cs.ucla.edu

**Quanquan Gu**
Department of Computer Science
University of California, Los Angeles
Los Angeles, CA 90095, USA
qgu@cs.ucla.edu

**Tong Zhang**
Dept. of Computer Science & Mathematics
Hong Kong Univ. of Science & Technology
Hong Kong, China
tongzhang@tongzhang-ml.org

## Abstract

A recent breakthrough in deep learning theory shows that the training of over-parameterized deep neural networks can be characterized by a kernel function called *neural tangent kernel* (NTK). However, it is known that this type of results does not perfectly match the practice, as NTK-based analysis requires the network weights to stay very close to their initialization throughout training, and cannot handle regularizers or gradient noises. In this paper, we provide a generalized neural tangent kernel analysis and show that noisy gradient descent with weight decay can still exhibit a "kernel-like" behavior. This implies that the training loss converges linearly up to a certain accuracy. We also establish a novel generalization error bound for two-layer neural networks trained by noisy gradient descent with weight decay.

## 1 Introduction

Deep learning has achieved tremendous practical success in a wide range of machine learning tasks [21, 19, 34]. However, due to the nonconvex and over-parameterized nature of modern neural networks, the success of deep learning cannot be fully explained by conventional optimization and machine learning theory.

A recent line of work studies the learning of over-parameterized neural networks in the so-called "neural tangent kernel (NTK) regime" [20]. It has been shown that the training of over-parameterized deep neural networks can be characterized by the training dynamics of kernel regression with the *neural tangent kernel* (NTK). Based on this, fast convergence rates can be proved for over-parameterized neural networks trained with randomly initialized (stochastic) gradient descent [16, 2, 15, 39, 40]. Moreover, it has also been shown that target functions in the NTK-induced reproducing kernel Hilbert space (RKHS) can be learned by wide enough neural networks with good generalization error [3, 4, 10].

Despite having beautiful theoretical results, the NTK-based results are known to have their limitations, for not perfectly matching the empirical observations in many aspects. Specifically, NTK-based analysis requires that the network weights stay very close to their initialization in the "node-wise" $\ell_2$ distance throughout the training. Moreover, due to this requirement, NTK-based analysis cannot handle regularizers such as weight decay, or large additive noises in the noisy gradient descent.

Given the advantages and disadvantages of the existing NTK-based results, a natural question is:

*Is it possible to establish the NTK-type results under more general settings?*

In this paper, we give an affirmative answer to this question by utilizing a mean-field analysis [12, 28, 27, 37, 17] to study neural tangent kernel. We show that with appropriate scaling, two-layer neural networks trained with noisy gradient descent and weight decay can still enjoy the nice theoretical guarantees.

We summarize the contributions of our paper as follows:

- Our analysis demonstrates that neural network training with noisy gradient and appropriate regularizers can still exhibit similar training dynamics as kernel methods, which is considered intractable in the neural tangent kernel literature, as the regularizer can easily push the network parameters far away from the initialization. Our analysis overcomes this technical barrier by relaxing the requirement on the closeness in the parameter space to the closeness in the distribution space. A direct consequence of our analysis is the linear convergence of noisy gradient descent up to certain accuracy for regularized neural network training. [1]

- We establish generalization bounds for the neural networks trained with noisy gradient descent with weight decay regularization. Our result shows that the infinitely wide neural networks trained by noisy gradient descent with weight decay can learn a class of functions that are defined based on a bounded $\chi^2$-divergence to initialization distribution. Different from standard NTK-type generalization bounds [1, 3, 10], our result can handle explicit regularization. Moreover, our proof is based on an extension of the proof technique in Meir and Zhang [29] from discrete distributions to continuous distributions, which may be of independent interest.

**Notation** We use lower case letters to denote scalars, and use lower and upper case bold face letters to denote vectors and matrices respectively. For a vector $\mathbf{x} = (x_1, \ldots, x_d)^\top \in \mathbb{R}^d$, and any positive integer $p$, we denote the $\ell_p$ norm of $\mathbf{x}$ as $\|\mathbf{x}\|_p = \left( \sum_{i=1}^d |x_i|^p \right)^{1/p}$. For a matrix $\mathbf{A} = (A_{ij}) \in \mathbb{R}^{m \times n}$, we denote by $\|\mathbf{A}\|_2$ and $\|\mathbf{A}\|_F$ its spectral and Frobenius norms respectively. We also define $\|\mathbf{A}\|_{\infty,\infty} = \max\{|A_{ij}| : 1 \leq i \leq m, 1 \leq j \leq n\}$. For a positive semi-definite matrix $\mathbf{A}$, we use $\lambda_{\min}(\mathbf{A})$ to denote its smallest eigenvalue.

For a positive integer $n$, we denote $[n] = \{1, \ldots, n\}$. We also use the following asymptotic notations. For two sequences $\{a_n\}$ and $\{b_n\}$, we write $a_n = O(b_n)$ if there exists an absolute constant $C$ such that $a_n \leq C b_n$. We also introduce $\widetilde{O}(\cdot)$ to hide the logarithmic terms in the Big-O notations.

At last, for two distributions $p$ and $p'$, we define the Kullback–Leibler divergence (KL-divergence) and $\chi^2$-divergence between $p$ and $p'$ as follows:

$$D_{\mathrm{KL}}(p\|p') = \int p(\mathbf{z}) \log \frac{p(\mathbf{z})}{p'(\mathbf{z})} d\mathbf{z}, \ \ D_{\chi^2}(p\|p') = \int \left( \frac{p(\mathbf{z})}{p'(\mathbf{z})} - 1 \right)^2 p'(\mathbf{z}) d\mathbf{z}.$$

## 2 Related Work

Our work is motivated by the recent study of neural network training in the "neural tangent kernel regime". In particular, Jacot et al. [20] first introduced the concept of neural tangent kernel by studying the training dynamics of neural networks with square loss. Based on neural tangent kernel, Allen-Zhu et al. [2], Du et al. [15], Zou et al. [39] proved the global convergence of (stochastic) gradient descent under various settings. Such convergence is later studied by a line of work [40] with improved network width conditions in various settings. Su and Yang [35], Cao et al. [9] studied the convergence along different eigendirections of the NTK. Chizat et al. [13] extended the similar idea to a more general framework called "lazy training". Liu et al. [26] studied the optimization for over-parameterized systems of non-linear equations. Allen-Zhu et al. [1], Arora et al. [3], Cao and Gu [11, 10] established generalization bounds for over-parameterized neural networks trained by (stochastic) gradient descent. Li et al. [25] studied noisy gradient descent with a certain learning rate schedule for a toy example.

**Algorithm 1** Noisy Gradient Descent for Training Two-layer Networks

---

**Input:** Step size $\eta$, total number of iterations $T$
Initialize $(\boldsymbol{\theta}_j, u_j) \sim p_0(\boldsymbol{\theta}, u)$, $j \in [m]$.
**for** $t = 0$ **to** $T - 1$ **do**
    Draw Gaussian noises $\zeta_{u,j} \sim N(0, 2\eta)$, $j \in [m]$
    $u_{t+1,j} = u_{t,j} - \eta \nabla_u \widehat{Q}(\{(\boldsymbol{\theta}_{t,j}, u_{t,j})\}_{j=1}^m) - \sqrt{\lambda}\zeta_{u,j}$
    Draw Gaussian noises $\boldsymbol{\zeta}_{\boldsymbol{\theta},j} \sim N(0, 2\eta \mathbf{I}_d)$, $j \in [m]$
    $\boldsymbol{\theta}_{t+1,j} = \boldsymbol{\theta}_{t,j} - \eta \nabla_{\boldsymbol{\theta}} \widehat{Q}(\{(\boldsymbol{\theta}_{t,j}, u_{t,j})\}_{j=1}^m) - \sqrt{\lambda}\boldsymbol{\zeta}_{\boldsymbol{\theta},j}$
**end for**

---

Our analysis follows the mean-field framework adopted in the recent line of work [5, 12, 28, 27, 37, 17, 18]. Bach [5] studied the generalization performance of infinitely wide two-layer neural networks under the mean-field setting. Chizat and Bach [12] showed the convergence of gradient descent for training infinitely wide, two-layer networks under certain structural assumptions. Mei et al. [28] proved the global convergence of noisy stochastic gradient descent and established approximation bounds between finite and infinite neural networks. Mei et al. [27] further showed that this approximation error can be independent of the input dimension in certain cases, and proved that under certain scaling condition, the residual dynamics of noiseless gradient descent is close to the dynamics of NTK-based kernel regression within certain bounded time interval $[0, T]$. Wei et al. [37] proved the convergence of a certain perturbed Wasserstein gradient flow, and established a generalization bound of the global minimizer of weakly regularized logistic loss. Fang et al. [17, 18] proposed a new concept called neural feature repopulation and extended the mean-field analysis.

## 3 Problem Setting and Preliminaries

In this section we introduce the basic problem setting for training an infinitely wide two-layer neural network, and explain its connection to the training dynamics of finitely wide neural networks.

Inspired by the study in Chizat et al. [13], Mei et al. [27], we introduce a scaling factor $\alpha > 0$ and study two-layer, infinitely wide neural networks of the form

$$f(p, \mathbf{x}) = \alpha \int_{\mathbb{R}^{d+1}} uh(\boldsymbol{\theta}, \mathbf{x})p(\boldsymbol{\theta}, u)d\boldsymbol{\theta}du, \tag{3.1}$$

where $\mathbf{x} \in \mathbb{R}^d$ is the input, $\boldsymbol{\theta} \in \mathbb{R}^d$ and $u \in \mathbb{R}$ are the first and second layer parameters respectively, $p(\boldsymbol{\theta}, u)$ is their joint distribution, and $h(\boldsymbol{\theta}, \mathbf{x})$ is the activation function. It is easy to see that (3.1) is the infinite-width limit of the following neural network of finite width

$$f_m(\{(\boldsymbol{\theta}_j, u_j)\}_{j=1}^m, \mathbf{x}) = \frac{\alpha}{m} \sum_{j=1}^m u_j h(\boldsymbol{\theta}_j, \mathbf{x}), \tag{3.2}$$

where $m$ is the number of hidden nodes, $\{(\boldsymbol{\theta}_j, u_j)\}_{j=1}^m$ are i.i.d. samples drawn from $p(\boldsymbol{\theta}, u)$. Note that choosing $\alpha = \sqrt{m}$ in (3.2) recovers the standard scaling in the neural tangent kernel regime [16], and setting $\alpha = 1$ in (3.1) gives the standard setting for mean-field analysis [28, 27].

We consider training the neural network with square loss and weight decay regularization. Let $S = \{(\mathbf{x}_1, y_1), \ldots, (\mathbf{x}_n, y_n)\}$ be the training data set, and $\phi(y', y) = (y' - y)^2$ be the square loss function. We consider Gaussian initialization $p_0(\boldsymbol{\theta}, u) \propto \exp[-u^2/2 - \|\boldsymbol{\theta}\|_2^2/2]$. Then for finite-width neural network (3.2), we define the training objective function as

$$\widehat{Q}(\{(\boldsymbol{\theta}_j, u_j)\}_{j=1}^m) = \mathbb{E}_S[\phi(f_m(\{(\boldsymbol{\theta}_j, u_j)\}_{j=1}^m, \mathbf{x}), y)] + \frac{\lambda}{m} \sum_{j=1}^m \left(\frac{u_j^2}{2} + \frac{\|\boldsymbol{\theta}_j\|_2^2}{2}\right), \tag{3.3}$$

where $\mathbb{E}_S[\cdot]$ denotes the empirical average over the training sample $S$, and $\lambda > 0$ is a regularization parameter. It is worth noting that when the network is wide enough, the neural network training is in the "interpolation regime", which gives zero training loss (first term in (3.3)). Therefore, even with a very large scaling parameter $\alpha$, weight decay (second term in (3.3)) is still effective.

In order to minimize the objective function $\widehat{Q}(\{(\boldsymbol{\theta}_j, u_j)\}_{j=1}^m)$ for the finite-width neural network (3.2), we consider the noisy gradient descent algorithm, which is displayed in Algorithm 1. It has been extensively studied [28, 12, 27, 17] in the mean-field regime that, the continuous-time, infinite-width

limit of Algorithm 1 can be characterized by the following partial differential equation (PDE) of the distribution $p_t(\boldsymbol{\theta}, u)^2$:

$$\frac{dp_t(\boldsymbol{\theta}, u)}{dt} = -\nabla_u[p_t(\boldsymbol{\theta}, u)g_1(t, \boldsymbol{\theta}, u)] - \nabla_{\boldsymbol{\theta}} \cdot [p_t(\boldsymbol{\theta}, u)g_2(t, \boldsymbol{\theta}, u)] + \lambda\Delta[p_t(\boldsymbol{\theta}, u)], \quad (3.4)$$

where

$$g_1(t, \boldsymbol{\theta}, u) = -\alpha\mathbb{E}_S[\nabla_{y'}\phi(f(p_t, \mathbf{x}), y)h(\boldsymbol{\theta}, \mathbf{x})] - \lambda u,$$
$$g_2(t, \boldsymbol{\theta}, u) = -\alpha\mathbb{E}_S[\nabla_{y'}\phi(f(p_t, \mathbf{x}), y)u\nabla_{\boldsymbol{\theta}}h(\boldsymbol{\theta}, \mathbf{x})] - \lambda\boldsymbol{\theta}.$$

Below we give an informal proposition to describe the connection between Algorithm 1 and the PDE (3.4). One can refer to Mei et al. [28], Chizat and Bach [12], Mei et al. [27] for more details on such approximation results.

**Proposition 3.1** (informal). Suppose that $h(\boldsymbol{\theta}, \mathbf{x})$ is sufficiently smooth, and PDE (3.4) has a unique solution $p_t$. Let $\{(\boldsymbol{\theta}_{t,j}, u_{t,j})\}_{j=1}^m$, $t \geq 0$ be output by Algorithm 1. Then for any $t \geq 0$ and any $\mathbf{x}$, it holds that

$$\lim_{m\to\infty}\lim_{\eta\to 0} f_m(\{(\boldsymbol{\theta}_{\lfloor t/\eta\rfloor, j}, u_{\lfloor t/\eta\rfloor, j})\}_{j=1}^m, \mathbf{x}) = f(p_t, \mathbf{x}).$$

Based on Proposition 3.1, one can convert the original optimization dynamics in the parameter space to the distributional dynamics in the probability measure space. In the rest of our paper, we mainly focus on $p_t(\boldsymbol{\theta}, u)$ defined by the PDE (3.4). It is worth noting that PDE (3.4) minimizes the following energy functional

$$Q(p) = L(p) + \lambda D_{\mathrm{KL}}(p||p_0), \quad (3.5)$$

where $L(p) = \mathbb{E}_S[\phi\big((f(p, \mathbf{x}), y)\big]$ is the empirical square loss, and $D_{\mathrm{KL}}(p||p_0) = \int p\log(p/p_0)d\boldsymbol{\theta}du$ is the KL-divergence between $p$ and $p_0$ [17]. The asymptotic convergence of PDE (3.4) towards the global minimum of (3.5) is recently established [28, 12, 27, 17].

Recall that, compared with the standard mean-field analysis, we consider the setting with an additional scaling factor $\alpha$ in (3.1). When $\alpha$ is large, we expect to build a connection to the recent results in the "neural tangent kernel regime" [27, 13], where the neural network training is similar to kernel regression using the neural tangent kernel $K(\mathbf{x}, \mathbf{x}')$ defined as $K(\mathbf{x}, \mathbf{x}') = K_1(\mathbf{x}, \mathbf{x}') + K_2(\mathbf{x}, \mathbf{x}')$, where

$$K_1(\mathbf{x}, \mathbf{x}') = \int u^2\langle\nabla_{\boldsymbol{\theta}}h(\boldsymbol{\theta}, \mathbf{x}), \nabla_{\boldsymbol{\theta}}h(\boldsymbol{\theta}, \mathbf{x}')\rangle p_0(\boldsymbol{\theta}, u)d\boldsymbol{\theta}du,$$

$$K_2(\mathbf{x}, \mathbf{x}') = \int h(\boldsymbol{\theta}, \mathbf{x})h(\boldsymbol{\theta}, \mathbf{x}')p_0(\boldsymbol{\theta}, u)d\boldsymbol{\theta}du.$$

Note that the neural tangent kernel function $K(\mathbf{x}, \mathbf{x}')$ is defined based on the initialization distribution $p_0$. This is because the specific network scaling in the neural tangent kernel regime forces the network parameters to stay close to initialization. In our analysis, we extend the definition of neural tangent kernel function to any distribution $p$, and define the corresponding Gram matrix $\mathbf{H} \in \mathbb{R}^{n\times n}$ of the kernel function on the training sample $S$ as follows:

$$\mathbf{H}(p) = \mathbf{H}_1(p) + \mathbf{H}_2(p), \quad (3.6)$$

where $\mathbf{H}_1(p)_{i,j} = \mathbb{E}_p[u^2\langle\nabla_{\boldsymbol{\theta}}h(\boldsymbol{\theta}, \mathbf{x}_i), \nabla_{\boldsymbol{\theta}}h(\boldsymbol{\theta}, \mathbf{x}_j)\rangle]$ and $\mathbf{H}_2(p)_{i,j} = \mathbb{E}_p[h(\boldsymbol{\theta}, \mathbf{x}_i)h(\boldsymbol{\theta}, \mathbf{x}_j)]$. Note that our definition of the Gram matrix $\mathbf{H}$ is consistent with a similar definition in Mei et al. [27].

Our study of NTK from the distributional perspective is based on the formulation of energy functional (3.5), which is different from the standard NTK analysis in the parameter space. Standard NTK analysis in the parameter space highly relies on the closeness of the parameters to the initialization. However, weight decay regularizer will push the global minima of the regularized loss to be close to the origin rather than the initialization. In addition, the gradient noises will also push the parameters towards random directions rather than the initialization. Therefore, the closeness to initialization in the parameter space no longer holds due to weight decay and gradient noises, and standard NTK analysis is not applicable anymore. To overcome this problem, we take a distributional approach, and show that with weight decay and gradient noises, the closeness to initialization in the distribution space holds. This enables us to carry out a generalized NTK analysis.

# 4 Main Results

In this section we present our main results on the optimization and generalization of infinitely wide two-layer neural networks trained with noisy gradient descent in Algorithm 1.

We first introduce the following two assumptions, which are required in both optimization and generalization error analyses.

**Assumption 4.1.** The data inputs and responses are bounded: $\|\mathbf{x}_i\|_2 \leq 1$, $|y_i| \leq 1$ for all $i \in [n]$.

Assumption 4.1 is a natural and mild assumption. Note that this assumption is much milder than the commonly used assumption $\|\mathbf{x}_i\|_2 = 1$ in the neural tangent kernel literature [16, 2, 39]. We would also like to remark that the bound 1 is not essential, and can be replaced by any positive constant.

**Assumption 4.2.** The activation function has the form $h(\boldsymbol{\theta}, \mathbf{x}) = \widetilde{h}(\boldsymbol{\theta}^\top \mathbf{x})$, where $\widetilde{h}(\cdot)$ is a three-times differentiable function that satisfies the following smoothness properties:

$$|\widetilde{h}(z)| \leq G_1, \quad |\widetilde{h}'(z)| \leq G_2, \quad |\widetilde{h}''(z)| \leq G_3, \quad |(z\widetilde{h}'(z))'| \leq G_4, \quad |\widetilde{h}'''(z)| \leq G_5,$$

where $G_1, \ldots, G_5$ are absolute constants, and we set $G = \max\{G_1, \ldots, G_5\}$ to simplify the bound.

$h(\boldsymbol{\theta}, \mathbf{x}) = \widetilde{h}(\boldsymbol{\theta}^\top \mathbf{x})$ is of the standard form in practical neural networks, and similar smoothness assumptions on $\widetilde{h}(\cdot)$ are standard in the mean field literature [28, 27]. Assumption 4.2 is satisfied by many smooth activation functions including the sigmoid and hyper-tangent functions.

## 4.1 Optimization Guarantees

In order to characterize the optimization dynamics defined by PDE (3.4), we need the following additional assumption.

**Assumption 4.3.** The Gram matrix of the neural tangent kernel defined in (3.6) is positive definite: $\lambda_{\min}(\mathbf{H}(p_0)) = \Lambda > 0$.

Assumption 4.3 is a rather weak assumption. In fact, Jacot et al. [20] has shown that if $\|\mathbf{x}_i\|_2 = 1$ for all $i \in [n]$, Assumption 4.3 holds as long as each pair of training inputs $\mathbf{x}_1, \ldots, \mathbf{x}_n$ are not parallel.

Now we are ready to present our main result on the training dynamics of infinitely wide neural networks.

**Theorem 4.4.** Let $\lambda_0 = \sqrt{\Lambda/n}$ and suppose that PDE (3.4) has a unique solution $p_t$. Under Assumptions 4.1, 4.2 and 4.3, if

$$\alpha \geq 8\sqrt{A_2^2 + \lambda A_1^2} \cdot \lambda_0^{-2} R^{-1}, \tag{4.1}$$

where $R = \min\left\{\sqrt{d+1}, [\text{poly}(G, \log(1/\lambda_0))]^{-1}\lambda_0^2\right\}$, then for all $t \in [0, +\infty)$, the following result hold:

$$L(p_t) \leq 2\exp(-2\alpha^2\lambda_0^2 t) + 2A_1^2\lambda^2\alpha^{-2}\lambda_0^{-4},$$
$$D_{\text{KL}}(p_t\|p_0) \leq 4A_2^2\alpha^{-2}\lambda_0^{-4} + 4A_1^2\lambda\alpha^{-2}\lambda_0^{-4},$$

where $A_1 = 2G(d+1) + 4G\sqrt{d+1}$ and $A_2 = 16G\sqrt{d+1} + 4G$.

Theorem 4.4 shows that the loss of the neural network converges linearly up to $O(\lambda^2\lambda_0^{-4}\alpha^{-2})$ accuracy, and the convergence rate depends on the smallest eigenvalue of the NTK Gram matrix. This matches the results for square loss in the neural tangent kernel regime [16]. However, we would like to emphasize that the algorithm we study here is noisy gradient descent, and the objective function involves a weight decay regularizer, both of which cannot be handled by the standard technical tools used in the NTK regime [2, 16, 15, 40]. Theorem 4.4 also shows that the KL-divergence between $p_t$ and $p_0$ is bounded and decreases as $\alpha$ increases. This is analogous to the standard NTK results [16, 2, 39] where the Euclidean distance between the parameter returned by (stochastic) gradient descent and its initialization is bounded, and decreases as the network width increases. The condition (4.1) requires a sufficiently large scaling factor $\alpha$, which is also analogous to the large scaling requirement in standard NTK analysis. It has been shown in [32, 40] that as long as the training data inputs are non-parallel to each other, $\lambda_{\min}(\mathbf{H}(p_0)) = \Omega(n^{-2})$. Therefore for non-parallel data, Theorem 4.4 requires $\alpha = \widetilde{\Omega}(n^3 d)$.

The results in Theorem 4.4 can also be compared with an earlier attempt by Mei et al. [27], which uses mean-field analysis to explain NTK. While Mei et al. [27] only reproduces the NTK-type results without regularization, our result holds for a more general setting with weight decay and noisy gradient. Another work by Tzen and Raginsky [36] uses mean-field analysis to study the lazy training of two-layer network. They consider a very small variance in parameter initialization, which is quite different from the practice of neural network training. In contrast, our work uses standard random initialization, and exactly follows the lazy training setting with scaling factor $\alpha$ [13]. Moreover, Tzen and Raginsky [36] only characterize the properties of the optimal solution without finite-time convergence result, while we characterize the whole training process with a linear convergence rate.

## 4.2 Generalization Bounds

Next, we study the generalization performance of the neural network obtained by minimizing the energy functional $Q(p)$. For simplicity, we consider the binary classification problem, and use the 0-1 loss $\ell^{0\text{-}1}(y', y) := \mathbb{1}\{y'y < 0\}$ to quantify the errors of the network, where $\mathbb{1}\{\cdot\}$ denotes the indicator function.

The following theorem presents the generalization bound for neural networks trained by Algorithm 1.

**Theorem 4.5.** Suppose that the training data $\{(\mathbf{x}_i, y_i)\}_{i=1}^n$ are i.i.d. sampled from an unknown but fixed distribution $\mathcal{D}$, and there exists a true distribution $p_{\text{true}}$ with $D_{\chi^2}(p_{\text{true}}||p_0) < \infty$, such that

$$y = \int u h(\boldsymbol{\theta}, \mathbf{x}) p_{\text{true}}(\boldsymbol{\theta}, u) d\boldsymbol{\theta} du$$

for all $(\mathbf{x}, y) \in \text{supp}(\mathcal{D})$. Let $p^*$ be the minimizer of the energy functional (3.5). Under Assumptions 4.1 and 4.2, if $\alpha \geq \sqrt{n\lambda} > 0$, then for any $\delta > 0$, with probability at least $1 - \delta$,

$$\mathbb{E}_{\mathcal{D}}[\ell^{0\text{-}1}(f(p^*, \mathbf{x}), y)] \leq (8G + 1)\sqrt{\frac{D_{\chi^2}(p_{\text{true}}||p_0)}{n}} + 6\sqrt{\frac{\log(2/\delta)}{2n}}.$$

Theorem 4.5 gives the generalization bound for the global minimizer of the energy functional $Q(p)$ obtained by noisy gradient descent with weight decay. The assumption on $p_{\text{true}}$ in Theorem 4.5 essentially assumes that the target function is in the function class $\mathcal{F} = \{f(\mathbf{x}) = \int u h(\boldsymbol{\theta}, \mathbf{x}) p_{\text{true}}(\boldsymbol{\theta}, u) d\boldsymbol{\theta} du, D_{\chi^2}(p_{\text{true}}||p_0) < +\infty\}$. We can see that the generalization bound gives a standard $1/\sqrt{n}$ error rate as long as $p_{\text{true}}$ has a constant $\chi^2$-divergence to $p_0$. Moreover, the $\chi^2$-divergence $D_{\chi^2}(p_{\text{true}}||p_0)$ also quantifies the difficulty for a target function defined by $p_{\text{true}}$ to be learnt. The larger $D_{\chi^2}(p_{\text{true}}||p_0)$ is, the more examples are needed to achieve the same target expected error.

Our generalization bound is different from existing NTK-based generalization results [24, 1, 3, 10], which highly rely on the fact that the learned neural network weights are close to the initialization. Therefore, these generalization error bounds no longer hold with the presence of regularizer and are not applicable to our setting. In addition, Bach [5] studied the generalization bounds for two-layer homogeneous networks and their connection to the NTK-induced RKHS.[3] Our result based on the KL-divergence regularization is different from their setting and is not covered by their results.

# 5 Proof Sketch of the Main Results

In this section we present a proof sketch for Theorems 4.4 and 4.5.

## 5.1 Proof Sketch of Theorem 4.4

We first introduce the following definition of 2-Wasserstein distance. For two distributions $p$ and $p'$ over $\mathbb{R}^{d+1}$, we define

$$\mathcal{W}_2(p, p') = \left( \inf_{\gamma \in \Gamma(p, p')} \int_{\mathbb{R}^{d+1} \times \mathbb{R}^{d+1}} \|\mathbf{z} - \mathbf{z}'\|_2^2 d\gamma(\mathbf{z}, \mathbf{z}') \right)^{1/2},$$

where $\Gamma(p, p')$ denotes the collection of all measures on $\mathbb{R}^d \times \mathbb{R}^d$ with marginals $p$ and $p'$ on the first and second factors respectively.

We also introduce the perturbation region $\mathcal{B}(p_0, R) := \{p | \mathcal{W}_2(p, p_0) \leq R\}$ based on the Wasserstein distance to the initialization, where $R$ defined in Theorem 4.4 gives the perturbation radius. We would like to highlight that compared with standard NTK-based analyses [1, 3, 10] which are based on a perturbation region around initial weight parameter, our proof is based upon the 2-Wasserstein neighborhood around $p_0$. Such an extension is essential to handle weight decay and gradient noises and is one of our key technical contributions.

The proof of Theorem 4.5 can be divided into the following three steps.

**Step 1: Landscape properties when $p_t$ is close to $p_0$.** We first consider the situation when the distribution $p_t$ is close to initial distribution $p_0$.

**Lemma 5.1.** Under Assumptions 4.1, 4.2 and 4.3, for any distribution $p$ with $\mathcal{W}_2(p, p_0) \leq R$, we have $\lambda_{\min}(\mathbf{H}(p)) \geq \Lambda/2$, where $R$ is defined in Theorem 4.4.

Lemma 5.1 shows that when $p_t$ close to $p_0$ in 2-Wasserstein distance, the Gram matrix at $p_t$ is strictly positive definite. This further implies nice landscape properties around $p_t$, which enables our analysis in the next step.

**Step 2: Loss and regularization bounds when $p_t$ is close to $p_0$.** With the results in Step 1, we establish loss and regularization bounds when $p_t$ stays in $\mathcal{B}(p_0, R)$ for some time period $[0, t^*]$, with $t^* = \inf\{t \geq 0 : \mathcal{W}_2(p_t, p_0) > R\}$. We have $t^* = +\infty$ if the $\{t \geq 0 : \mathcal{W}_2(p_t, p_0) > R\} = \emptyset$. The following lemma shows that the loss function decreases linearly in the time perioud $[0, t^*]$.

**Lemma 5.2.** Under Assumptions 4.1, 4.2 and 4.3, for any $t \leq t^*$, it holds that

$$\sqrt{L(p_t)} \leq \exp(-\alpha^2 \lambda_0^2 t) + A_1 \lambda \alpha^{-1} \lambda_0^{-2},$$

where $A_1$ is defined in Theorem 4.4.

Besides the bound on $L(p_t)$, we also have a bound on the KL-divergence between $p_t$ and $p_0$, as is given in the following lemma.

**Lemma 5.3.** Under Assumptions 4.1, 4.2 and 4.3, for any $t \leq t^*$,

$$D_{\mathrm{KL}}(p_t || p_0) \leq 4 A_2^2 \alpha^{-2} \lambda_0^{-4} + 4 A_1^2 \lambda \alpha^{-2} \lambda_0^{-4},$$

where $A_1$ and $A_2$ are defined in Theorem 4.4.

Here we would like to remark that the bound in Lemma 5.3 does not increase with time $t$, which is an important feature of our result. This is achieved by jointly considering two types of bounds on the KL-divergence between $p_t$ and $p_0$: the first type of bound is on the time derivative of $D_{\mathrm{KL}}(p_t, p_0)$ based on the training dynamics described by (3.4), and the second type of bound is a direct KL-divergence bound based on the monotonicity of the energy functional $Q(p_t)$. The detailed proof of this lemma is deferred to the appendix.

**Step 3: Large scaling factor $\alpha$ ensures distribution closeness throughout training.** When $\alpha$ is sufficiently large, $p_t$ will not escape from the perturbation region. To show this, we utilize the following Talagrand inequality (see Corollary 2.1 in Otto and Villani [31] and Theorem 9.1.6 in Bakry et al. [6]), which is based on the fact that in our setting $p_0$ is a Gaussian distribution.

**Lemma 5.4** (Otto and Villani [31])**.** The probability measure $p_0(\boldsymbol{\theta}, u) \propto \exp[-u^2/2 - \|\boldsymbol{\theta}\|_2^2/2]$ satisfies following Talagrand inequality

$$\mathcal{W}_2(p, p_0) \leq 2\sqrt{D_{\mathrm{KL}}(p || p_0)}.$$

The main purpose of Lemma 5.4 is to build a connection between the 2-Wasserstein ball around $p_0$ and the KL-divergence ball.

We are now ready to finalize the proof. Note that given our results in Step 2, it suffices to show that $t^* = +\infty$, which is proved based on a reduction to absurdity.

*Proof of Theorem 4.4.* By the definition of $t^*$, for any $t \leq t^*$, we have

$$\mathcal{W}_2(p_t, p_0) \leq 2 D_{\mathrm{KL}}(p_t || p_0)^{1/2} \leq 2\big(4 A_2^2 \alpha^{-2} \lambda_0^{-4} + 4 A_1^2 \lambda \alpha^{-2} \lambda_0^{-4}\big)^{1/2} \leq R/2,$$

where the first inequality is by Lemma 5.4, the second inequality is by Lemma 5.3 ,and the third inequality is due to the choice of $\alpha$ in (4.1).

This deduces that the set $\{t \geq 0 : \mathcal{W}_2(p_t, p_0) > R\}$ is empty and $t^* = \infty$, because otherwise $\mathcal{W}_2(p_{t^*}, p_0) = R$ by the continuity of 2-Wasserstein distance. Therefore the results of Lemmas 5.2 and 5.3 hold for all $t \in [0, +\infty)$. Squaring both sides of the inequality in Lemma 5.2 and applying Jensen's inequality gives

$$L(p_t) \leq 2 \exp(-2\alpha^2 \lambda_0^2 t) + 2A_1^2 \lambda^2 \alpha^{-2} \lambda_0^{-4}.$$

This completes the proof. □

## 5.2 Proof Sketch of Theorem 4.5

For any $M > 0$, we consider the following class of infinitely wide neural network functions characterized by the KL-divergence to initialization

$$\mathcal{F}_{\mathrm{KL}}(M) = \{f(p, \mathbf{x}) : D_{\mathrm{KL}}(p\|p_0) \leq M\}. \tag{5.1}$$

Our proof consists of the following two steps.

**Step 1: A KL-divergence based Rademacher complexity bound.** Motivated by the KL-divergence regularization in the energy functional $Q(p)$, we first derive a Rademacher complexity bound for the function class $\mathcal{F}_{\mathrm{KL}}(M)$, which is given as the following lemma.

**Lemma 5.5.** Suppose that $|h(\boldsymbol{\theta}, \mathbf{x})| \leq G$ for all $\boldsymbol{\theta}$ and $\mathbf{x}$, and $M \leq 1/2$. Then

$$\mathfrak{R}_n(\mathcal{F}_{\mathrm{KL}}(M)) \leq 2G\alpha\sqrt{\frac{M}{n}}.$$

Lemma 5.5 is another key technical contribution of our paper. Different from previous NTK-based generalization error analysis that utilizes the approximation of neural network functions with linear models [22, 10], we use mean-field analysis to directly bound the Rademacher complexity of neural network function class. At the core of the proof for Lemma 5.5 is an extension of Meir and Zhang [29] for discrete distributions to continuous distributions, which is of independent interest.

**Step 2: Expected 0-1 loss bound over $\mathcal{F}_{\mathrm{KL}}(M)$.** We bound the expected 0-1 loss by the square root of the empirical square loss function and the Rademacher complexity.

**Lemma 5.6.** For any $\delta > 0$, with probability at least $1 - \delta$, the following bound holds uniformly over all $f \in \mathcal{F}_{\mathrm{KL}}(M)$:

$$\mathbb{E}_{\mathcal{D}}[\ell^{0\text{-}1}(f(\mathbf{x}), y)] \leq \sqrt{\mathbb{E}_S[|f(\mathbf{x}) - y|^2]} + 4\mathfrak{R}_n(\mathcal{F}_{\mathrm{KL}}(M)) + 6\sqrt{\frac{\log(2/\delta)}{2n}}.$$

The bound in Lemma 5.6 utilizes the property of square loss instead of margin-based arguments [7]. In this way, we are able to obtain a tighter bound, as our setting uses square loss as the objective.

We now finalize the proof by deriving the loss and regularization bounds at $p^*$ and plugging them into Lemma 5.6.

*Proof of Theorem 4.5.* Let $\overline{D} = D_{\chi^2}(p_{\mathrm{true}}\|p_0) < \infty$. Define

$$\widehat{p} = \frac{\alpha - 1}{\alpha} \cdot p_0 + \frac{1}{\alpha} \cdot p_{\mathrm{true}}.$$

Then we have $\int \widehat{p}(\theta, u) du d\theta = 1$, $\widehat{p}(\theta, u) \geq 0$, meaning that $\widehat{p}$ is a well-defined density function. The training loss of $\widehat{p}$ can be calculated as follows:

$$L(\widehat{p}) = \mathbb{E}_S\left[\alpha \int uh(\theta, \mathbf{x})\widehat{p}(u, \theta) du d\theta - y\right]^2 = \mathbb{E}_S\left(0 + \alpha \cdot \frac{1}{\alpha}y - y\right)^2 = 0. \tag{5.2}$$

Moreover, by the fact that KL-divergence is upper bounded by the $\chi^2$-divergence, we have

$$D_{\mathrm{KL}}(\widehat{p}\|p_0) \leq D_{\chi^2}(\widehat{p}\|p_0) = \int \left[\frac{\alpha - 1}{\alpha} + \frac{p_{\mathrm{true}}(\boldsymbol{\theta}, u)}{\alpha p_0(\boldsymbol{\theta}, u)} - 1\right]^2 p_0(\boldsymbol{\theta}, u) d\boldsymbol{\theta} du = \alpha^{-2}\overline{D}, \tag{5.3}$$

where the last equation is by the definition of $\chi^2$-divergence. Now we have
$$Q(p^*) \leq Q(\widehat{p}) = L(\widehat{p}) + \lambda D_{\mathrm{KL}}(\widehat{p}\|p_0) \leq \alpha^{-2}\lambda\overline{D},$$
where the first inequality follows by the optimality of $p^*$, and we plug (5.2), (5.3) into the definition of the energy function $Q(p)$ in (3.5) to obtain the second inequality. Applying the definition of $Q(p)$ again gives the following two bounds:
$$L(p^*) = \mathbb{E}_S[|f(p^*, \mathbf{x}) - y|^2] \leq \alpha^{-2}\lambda\overline{D}, \tag{5.4}$$
$$D_{\mathrm{KL}}(p^*\|p_0) \leq \alpha^{-2}\overline{D}. \tag{5.5}$$
By (5.5), we have $f(p^*, \mathbf{x}) \in \mathcal{F}_{\mathrm{KL}}(\alpha^{-2}\widehat{D})$. Therefore, applying Lemma 5.6 with $M = \alpha^{-2}\widehat{D}$ gives

$$\mathbb{E}_{\mathcal{D}}[\ell^{0\text{-}1}(f(p^*, \mathbf{x}), y)] \leq \sqrt{\mathbb{E}_S[|f(p^*, \mathbf{x}) - y|^2]} + 4\mathfrak{R}_n(\mathcal{F}_{\mathrm{KL}}(\alpha^{-2}\widehat{D})) + 6\sqrt{\frac{\log(2/\delta)}{2n}}$$

$$\leq \sqrt{\alpha^{-2}\lambda\overline{D}} + 8G\alpha\sqrt{\frac{\alpha^{-2}\widehat{D}}{n}} + 6\sqrt{\frac{\log(2/\delta)}{2n}}$$

$$\leq (8G + 1)\sqrt{\frac{\widehat{D}}{n}} + 6\sqrt{\frac{\log(2/\delta)}{2n}},$$

where the second inequality follows from (5.4) and Lemma 5.5, and the third inequality follows from the assumption that $\alpha \geq \sqrt{n\lambda}$. This finishes the proof. $\qquad\square$

## 6  Conclusion

In this paper, we demonstrate that the neural tangent kernel based regression can characterize neural network training dynamics in a general setting where weight decay and gradient noises are implemented. This leads to the linear convergence of noisy gradient descent up to certain accuracy. Compared with existing analysis in the neural tangent kernel regime, our work points out an important observation that as long as the distribution of parameters stays close to the initialization, it does not matter whether the parameters themselves are close to their initial values. We also establish a novel generalization bound for the neural network trained by noisy gradient descent with weight decay regularization.

## Broader Impact

Deep learning has achieved tremendous success in various real-world applications such as image recognition, natural language processing, self-driving cars and disease diagnosis. However, many deep learning models are not interpretable, which greatly limits their application and can even cause danger in safety-critical applications. This work aims to theoretically explain the success of learning neural networks, and can help add transparency to deep learning methods that have been implemented and deployed in real applications. Our result makes deep learning more interpretable, which is crucial in applications such as self-driving cars and disease diagnosis. Moreover, our results can potentially guide the design of new deep learning models with better performance guarantees.

As a paper focusing on theoretical results, no risk can be directly caused. However, if the theoretical results are over-interpreted and blindly used to design deep learning models for specific applications, bad performance may be expected as there is still some gap between theory and practice.

## Acknowledgement

We would like to thank the anonymous reviewers for their helpful comments. ZC, YC and QG are partially supported by the National Science Foundation CAREER Award 1906169, IIS-2008981 and Salesforce Deep Learning Research Award. TZ is supported by GRF 16201320. The views and conclusions contained in this paper are those of the authors and should not be interpreted as representing any funding agencies.

## Footnotes

[1]Although we focus on the continuous-time limit of the noisy gradient descent algorithm, our result can be extended to the discrete-time setting by applying the approximation results in Mei et al. [28]

[2]Throughout this paper, we define $\nabla$ and $\Delta$ without subscripts as the gradient/Laplacian operators with respect to the full parameter collection $(\boldsymbol{\theta}, u)$.

[3]Although it is not named "neural tangent kernel", the kernel function studied in [5] is essentially NTK.

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
