[Supplementary Material]

# A  Proof of Lemmas in Section 5

In this section we provide the proofs of lemmas we use in Section 5 for the proof of our main results. We first introduce the following notations. We denote $\mathbf{f}(t) = (f(p_t, \mathbf{x}_1), \ldots, f(p_t, \mathbf{x}_n))^\top$. Moreover, we define

$$\widehat{g}_1(t, \boldsymbol{\theta}, u) = -\alpha \mathbb{E}_S[\nabla_f \phi(f(p_t, \mathbf{x}), y) h(\boldsymbol{\theta}, \mathbf{x})], \tag{A.1}$$

$$\widehat{g}_2(t, \boldsymbol{\theta}, u) = -\alpha \mathbb{E}_S[\nabla_f \phi(f(p_t, \mathbf{x}), y) u \nabla_{\boldsymbol{\theta}} h(\boldsymbol{\theta}, \mathbf{x})]. \tag{A.2}$$

## A.1  Proof of Lemma 5.1

Here we give the proof of Lemma 5.1. The following lemma summarizes some basic properties of the activation function $h(\boldsymbol{\theta}, u)$.

**Lemma A.1.** Under Assumptions 4.1 and 4.2, for all $\mathbf{x}$ and $\boldsymbol{\theta}$, it holds that $|h(\boldsymbol{\theta}, \mathbf{x})| \leq G$, $\|\nabla_{\boldsymbol{\theta}} h(\boldsymbol{\theta}, \mathbf{x})\|_2 \leq G$, $|\Delta h(\boldsymbol{\theta}, \mathbf{x})| \leq G$, $\|\nabla_{\boldsymbol{\theta}} h(\boldsymbol{\theta}_1, x) - \nabla_{\boldsymbol{\theta}} h(\boldsymbol{\theta}_2, x)\|_2 \leq G\|\boldsymbol{\theta}_1 - \boldsymbol{\theta}_2\|_2$, $\|\nabla_{\boldsymbol{\theta}}(\nabla_{\boldsymbol{\theta}} h(\boldsymbol{\theta}, \mathbf{x}) \cdot \boldsymbol{\theta})\|_2 \leq G$, $\|\nabla_{\boldsymbol{\theta}} \Delta_{\boldsymbol{\theta}} h(\boldsymbol{\theta}, \mathbf{x})\|_2 \leq G$.

We also give the following two lemmas to characterize the difference between the Gram matrices defined with $p_0$ and some other distribution $p$ that is close to $p_0$ in 2-Wasserstein distance.

**Lemma A.2.** Under Assumptions 4.1 and 4.2, for any distribution $p$ with $\mathcal{W}_2(p, p_0) \leq \sqrt{d+1}$ and any $r > 0$,

$$\|\mathbf{H}_1(p) - \mathbf{H}_1(p_0)\|_{\infty,\infty} \leq G^2 \left[\sqrt{8d+10} + 2r^2 G^2\right] \mathcal{W}_2(p, p_0) + 2G^2 \mathbb{E}_{p_0}[u_0^2 \mathbb{1}(|u_0 \geq r|)].$$

**Lemma A.3.** Under Assumptions 4.1 and 4.2, for any distribution $p$ with $\mathcal{W}_2(p, p_0) \leq \sqrt{d+1}$,

$$\|\mathbf{H}_2(p) - \mathbf{H}_2(p_0)\|_{\infty,\infty} \leq 2G^2 \mathcal{W}_2(p, p_0).$$

The following lemma gives a tail bound with respect to our initialization distribution $p_0$, which we frequently utilize for truncation arguments.

**Lemma A.4.** The initialization distribution $p_0$ satisfies the following tail bound:

$$\mathbb{E}_{p_0}[u_0^2 \mathbb{1}(|u_0| \geq r)] \leq \frac{\exp(-r^2/4)}{2}.$$

We are now ready to provide the proof of Lemma 5.1.

*Proof of Lemma 5.1.*  Here we first give the definition of $R$ in Theorem 4.4 with specific polynomial dependencies.

$$R = \min\left\{\sqrt{d+1}, [\text{poly}(G, \log(n/\Lambda))n/\Lambda]^{-1}\right\}$$

$$\leq \min\left\{\sqrt{d+1}, \left(8G^2\sqrt{8d+10} + 64G^2 \log(8\Lambda^{-1} n G^2)\right)^{-1} n^{-1}\Lambda\right\}.$$

Note that the definition of $R$, the results for Lemmas A.2 and A.3 hold for all $p$ with $\mathcal{W}_2(p, p_0) \leq R$. Now by Lemma A.2, for any $p$ with $\mathcal{W}_2(p, p_0) \leq R$ and any $r > 0$,

$$\|\mathbf{H}_1(p) - \mathbf{H}_1(p_0)\|_{\infty,\infty} \leq G^2 R\sqrt{8d+10} + 2r^2 G^2 R + 2G^2 \mathbb{E}_{p_0}[u_0^2 \mathbb{1}(|u_0 \geq r|)]. \tag{A.3}$$

Choose $r = 2\sqrt{\log(8\Lambda^{-1} n G^2)}$, then by Lemma A.4 we have

$$\mathbb{E}_{p_0}[u_0^2 \mathbb{1}(|u_0 \geq r|)] \leq \frac{\Lambda}{16nG^2}. \tag{A.4}$$

Moreover, by the definition of $R$, we have

$$R \leq \left(8G^2\sqrt{8d+10} + 16G^2 r^2\right)^{-1} n^{-1}\Lambda. \tag{A.5}$$

Plugging the bounds on $\mathbb{E}_{p_0}[u_0^2 \mathbb{1}(|u_0 \geq r|)]$ and $R$ given by (A.4) and (A.5) into (A.3) gives

$$\|\mathbf{H}_1(p) - \mathbf{H}_1(p_0)\|_{\infty,\infty} \leq G^2 R \sqrt{8d + 10} + 2r^2 G^2 R + G^2 \mathbb{E}_{p_0}[u_0^2 \mathbb{1}(|u_0 \geq r|)] \tag{A.6}$$

$$\leq \frac{\Lambda}{8n} + \frac{\Lambda}{8n} \tag{A.7}$$

$$= \frac{\Lambda}{4n}. \tag{A.8}$$

By Lemma A.3, for any distribution $p$ with $\mathcal{W}_2(p, p_0) \leq R$,

$$\|\mathbf{H}_2(p) - \mathbf{H}_2(p_0)\|_{\infty,\infty} \leq 2G^2 R. \tag{A.9}$$

The definition of $R$ also leads to the following bound:

$$R \leq (8G^2)^{-1} n^{-1} \Lambda. \tag{A.10}$$

Therefore we can plug the bound (A.10) into (A.9), which gives

$$\|\mathbf{H}_2(p) - \mathbf{H}_2(p_0)\|_{\infty,\infty} \leq \frac{\Lambda}{4n}. \tag{A.11}$$

Combining (A.6) and (A.11) further gives

$$\|\mathbf{H}(p) - \mathbf{H}(p_0)\|_{\infty,\infty} \leq \|\mathbf{H}_1(p) - \mathbf{H}_1(p_0)\|_{\infty,\infty} + \|\mathbf{H}_2(p) - \mathbf{H}_2(p_0)\|_{\infty,\infty} \leq \frac{\Lambda}{2n}.$$

Then by standard matrix perturbation bounds, we have $\lambda_{\min}(\mathbf{H}(p)) \geq \lambda_{\min}(\mathbf{H}(p_0)) - \|\mathbf{H}(p) - \mathbf{H}(p_0)\|_2 \geq \lambda_{\min}(\mathbf{H}(p_0)) - n\|\mathbf{H}(p) - \mathbf{H}(p_0)\|_{\infty,\infty} \geq \Lambda/2$, which finishes the proof. $\square$

## A.2 Proof of Lemma 5.2

Here we give the proof of Lemma 5.2. The following lemma summarizes some basic calculation on the training dynamics. Here we remind the readers that the definitions of $\widehat{g}_1(t, \boldsymbol{\theta}, u)$ and $\widehat{g}_2(t, \boldsymbol{\theta}, u)$ are given in (A.1) and (A.2) respectively.

**Lemma A.5.** Let $p_t$ be the solution of PDE (3.4). Then the following identity holds.

$$\frac{\partial L(p_t)}{\partial t} = -\int_{\mathbb{R}^{d+1}} p_t(\boldsymbol{\theta}, u) \|\widehat{g}_1(t, \boldsymbol{\theta}, u)\|_2^2 d\boldsymbol{\theta} du - \int_{\mathbb{R}^{d+1}} p_t(\boldsymbol{\theta}, u) |\widehat{g}_2(t, \boldsymbol{\theta}, u)|^2 d\boldsymbol{\theta} du$$
$$+ \lambda \int_{\mathbb{R}^{d+1}} p_t(\boldsymbol{\theta}, u) [\widehat{g}_1 \cdot u + \widehat{g}_2 \cdot \boldsymbol{\theta} - \nabla_u \cdot \widehat{g}_1 - \nabla_{\boldsymbol{\theta}} \cdot \widehat{g}_2] d\boldsymbol{\theta} du. \tag{A.12}$$

Lemma A.5 decomposes the time derivative of $L(p_t)$ into several terms. The following two lemmas further provides bounds on these terms. Note that by the definition in (A.1) and (A.2), Lemma A.6 below essentially serves as a bound on the first two terms on the right-hand side of (A.12).

**Lemma A.6.** Under Assumptions 4.1, 4.2 and 4.3, let $\lambda_0$ be defined in Theorem 4.4. Then for $t \leq t^*$, it holds that

$$\int_{\mathbb{R}^{d+1}} p_t(\boldsymbol{\theta}, u) \big[ |\mathbb{E}_S[(f(p_t, \mathbf{x}) - y)h(\boldsymbol{\theta}, \mathbf{x})]|^2 + \|\mathbb{E}_S[(f(p_t, \mathbf{x}) - y)u\nabla_{\boldsymbol{\theta}} h(\boldsymbol{\theta}, \mathbf{x})]\|_2^2 \big] d\boldsymbol{\theta} du \geq \frac{\lambda_0^2}{2} L(p_t).$$

**Lemma A.7.** Under Assumptions 4.1 and 4.2, let $A_1$ be defined in Theorem 4.4. Then for $t \leq t^*$, it holds that

$$\int_{\mathbb{R}^{d+1}} p_t(\boldsymbol{\theta}, u) [\widehat{g}_1 \cdot u + \widehat{g}_2 \cdot \boldsymbol{\theta} - \nabla_u \cdot \widehat{g}_1 - \nabla_{\boldsymbol{\theta}} \cdot \widehat{g}_2] d\boldsymbol{\theta} du \leq 2\alpha A_1 \sqrt{L(p_t)}.$$

We now present the proof of Lemma 5.2, which is based on the calculations in Lemmas A.5, A.6 and A.7 as well as the application of Gronwall's inequality.

*Proof of Lemma 5.2.* By Lemma A.5, we have

$$\frac{\partial L(p_t)}{\partial t} = -\underbrace{\left[\int_{\mathbb{R}^{d+1}} p_t(\boldsymbol{\theta}, u)\|\widehat{g}_1(t, \boldsymbol{\theta}, u)\|_2^2 d\boldsymbol{\theta} du + \int_{\mathbb{R}^{d+1}} p_t(\boldsymbol{\theta}, u)|\widehat{g}_2(t, \boldsymbol{\theta}, u)|^2 d\boldsymbol{\theta} d\boldsymbol{\theta} du\right]}_{I_1}$$

$$+ \underbrace{\lambda \int_{\mathbb{R}^{d+1}} p_t(\boldsymbol{\theta}, u)[\widehat{g}_1 \cdot u + \widehat{g}_2 \cdot \boldsymbol{\theta} - \nabla \cdot \widehat{g}_1 - \nabla \cdot \widehat{g}_2] d\boldsymbol{\theta} du}_{I_2} ., \tag{A.13}$$

For $I_1$, we have

$$I_1 = 4\alpha^2 \int_{\mathbb{R}^{d+1}} p_t(\boldsymbol{\theta}, u)\big[|\mathbb{E}_S[(f(p_t, \mathbf{x}) - y)h(\boldsymbol{\theta}, \mathbf{x})]|^2 \tag{A.14}$$

$$+ \|\mathbb{E}_S[(f(p_t, \mathbf{x}) - y)u\nabla_{\boldsymbol{\theta}} h(\boldsymbol{\theta}, \mathbf{x})]\|_2^2\big] d\boldsymbol{\theta} du \tag{A.15}$$

$$\geq 2\alpha^2 \lambda_0^2 L(p_t), \tag{A.16}$$

where the equation follows by the definitions of $\widehat{g}_1(t, \boldsymbol{\theta}, u)$, $\widehat{g}_2(t, \boldsymbol{\theta}, u)$ in (A.1), (A.1), and the inequality follows by Lemma A.6. For $I_2$, we directly apply Lemma A.7 and obtain

$$I_2 \leq 2A_1 \alpha \lambda \sqrt{L(p_t)}. \tag{A.17}$$

Plugging the bounds (A.16) and (A.17) into (A.13) yields

$$\frac{\partial L(p_t)}{\partial t} \leq -2\alpha^2 \lambda_0^2 L(p_t) + 2A_1 \alpha \lambda \sqrt{L(p_t)}. \tag{A.18}$$

Now denote $V(t) = \sqrt{L(p_t)} - A_1 \lambda \alpha^{-1} \lambda_0^{-2}$. Then (A.18) implies that[4]

$$\frac{\partial V(t)}{\partial t} \leq -\alpha^2 \lambda_0^2 V(t).$$

By Gronwall's inequality we further get

$$V(t) \leq \exp(-\alpha^2 \lambda_0^2 t)V(0).$$

By $V(0) = \sqrt{L(p_0)} - A_1 \lambda \alpha^{-1} \lambda_0^{-2} \leq \sqrt{L(p_0)} \leq 1$, we have

$$\sqrt{L(p_t)} \leq \exp(-\alpha^2 \lambda_0^2 t) + A_1 \lambda \alpha^{-1} \lambda_0^{-2}. \tag{A.19}$$

This completes the proof. $\qquad\square$

### A.3 Proof of Lemma 5.3

In this subsection we present the proof of Lemma 5.3.

**Lemma A.8.** Under Assumptions 4.1, 4.2 and 4.3, let $\lambda_0$ be defined in Theorem 4.4. Then for $t \leq t^*$ the following inequality holds

$$D_{\mathrm{KL}}(p_t||p_0) \leq 2A_2^2 \alpha^{-2} \lambda_0^{-4} + 2A_2^2 A_1^2 \lambda^2 \lambda_0^{-4} t^2.$$

If $\lambda \neq 0$, the KL distance bound given by Lemma A.8 depends on $t$, we can give a tighter bound by the monotonically deceasing property of $Q(p_t)$ given by the following lemma, which states that the energy functional is monotonically decreasing during training. Note that this is not a new result, as it is to some extent an standard result, and has been discussed in Mei et al. [28, 27], Fang et al. [17].

**Lemma A.9.** Let $p_t$ be the solution of PDE (3.4). Then $Q(p_t)$ is monotonically deceasing, i.e.,

$$\frac{\partial Q(p_t)}{\partial t} \leq 0. \tag{A.20}$$

*Proof of Lemma 5.3.* Notice that for $\lambda = 0$, Lemma A.8 directly implies the conclusion. So in the rest of the proof we consider the situation where $\lambda > 0$. Denote $t_0 = A_1^{-1}\alpha^{-1}\lambda^{-1}$, we consider two cases $t_0 \geq t_*$ and $t_0 < t_*$ respectively.

If $t_0 \geq t_*$, then for $t \leq t^*$ we have $t \leq t_0$

$$
\begin{aligned}
D_{\mathrm{KL}}(p_t||p_0) &\leq 2A_2^2\alpha^{-2}\lambda_0^{-4} + 2A_2^2A_1^2\lambda^2\lambda_0^{-4}t^2 \\
&\leq 2A_2^2\alpha^{-2}\lambda_0^{-4} + 2A_2^2A_1^2\lambda^2\lambda_0^{-4}t_0^2 \\
&= 4A_2^2\alpha^{-2}\lambda_0^{-4} \\
&\leq 4A_2^2\alpha^{-2}\lambda_0^{-4} + 4A_1^2\lambda\alpha^{-2}\lambda_0^{-4},
\end{aligned}
$$

where the first inequality is by Lemma A.8 and the second inequality is by $t \leq t_0$.

If $t_0 < t_*$, then for $t \leq t_0$, we also have

$$
D_{\mathrm{KL}}(p_t||p_0) \leq 4A_2^2\alpha^{-2}\lambda_0^{-4} \leq 4A_2^2\alpha^{-2}\lambda_0^{-4} + 4A_1^2\lambda\alpha^{-2}\lambda_0^{-4}.
$$

For $t_0 < t \leq t_*$, consider $Q(p_t) = L(p_t) + \lambda D_{\mathrm{KL}}(p_t||p_0)$. The monotonically deceasing property of $Q(p_t)$ in Lemma A.9 implies that,

$$
D_{\mathrm{KL}}(p_t||p_0) \leq \lambda^{-1}Q(p_t) \leq \lambda^{-1}Q(p_{t_0}). \tag{A.21}
$$

Now we bound $Q(p_{t_0})$. We first bound $L(p_{t_0})$. Squaring both sides of the result of Lemma 5.2 and applying Jensen's inequality now gives

$$
L(p_t) \leq 2\exp(-2\alpha^2\lambda_0^2 t) + 2A_1^2\lambda^2\alpha^{-2}\lambda_0^{-4}. \tag{A.22}
$$

Plugging $t_0 = A_1^{-1}\alpha^{-1}\lambda^{-1}$ into (A.22) gives

$$
\begin{aligned}
L(p_{t_0}) &\leq 2\exp(-2\alpha^2\lambda_0^2 t_0) + 2A_1^2\lambda^2\alpha^{-2}\lambda_0^{-4} \\
&= 2\exp\left(-2A_1^{-1}\lambda^{-1}\alpha\lambda_0^2\right) + 2A_1^2\lambda^2\alpha^{-2}\lambda_0^{-4} \\
&\leq 4A_1^2\lambda^2\alpha^{-2}\lambda_0^{-4}, \tag{A.23}
\end{aligned}
$$

where the last inequality is by $\exp(-2z) = [\exp(-z)]^2 \leq [1/z]^2$ for any $z > 0$. We then bound $D_{\mathrm{KL}}(p_{t_0}||p_0)$. By Lemma A.8, we have

$$
D_{\mathrm{KL}}(p_{t_0}||p_0) \leq 2A_2^2\alpha^{-2}\lambda_0^{-4} + 2A_2^2A_1^2\lambda^2\lambda_0^{-4}t_0^2 = 4A_2^2\alpha^{-2}\lambda_0^{-4}. \tag{A.24}
$$

Plugging (A.23) and (A.24) into (A.21) gives

$$
D_{\mathrm{KL}}(p_t||p_0) \leq \lambda^{-1}Q(p_{t_0}) = \lambda^{-1}L(p_{t_0}) + D_{\mathrm{KL}}(p_{t_0}||p_0) \leq 4A_2^2\alpha^{-2}\lambda_0^{-4} + 4A_1^2\lambda\alpha^{-2}\lambda_0^{-4}.
$$

This completes the proof. $\qquad\square$

## A.4 Proof of Lemma 5.5

*Proof of Lemma 5.5.* Our proof is inspired by the Rademacher complexity bound for discrete distributions given by Meir and Zhang [29]. Let $\gamma$ be a parameter whose value will be determined later in the proof. We have

$$
\begin{aligned}
\mathfrak{R}_n(\mathcal{F}_{\mathrm{KL}}(M)) &= \frac{\alpha}{\gamma} \cdot \mathbb{E}_{\boldsymbol{\xi}}\left[\sup_{p:D_{\mathrm{KL}}(p||p_0)\leq M}\int_{\mathbb{R}^{d+1}}\frac{\gamma}{n}\sum_{i=1}^n\xi_i uh(\boldsymbol{\theta},\mathbf{x}_i)p(\boldsymbol{\theta},u)d\boldsymbol{\theta}du\right] \\
&\leq \frac{\alpha}{\gamma}\cdot\left\{M + \mathbb{E}_{\boldsymbol{\xi}}\log\left[\int\exp\left(\frac{\gamma}{n}\sum_{i=1}^n\xi_i uh(\boldsymbol{\theta},\mathbf{x}_i)\right)p_0(\boldsymbol{\theta},u)d\boldsymbol{\theta}du\right]\right\} \\
&\leq \frac{\alpha}{\gamma}\cdot\left\{M + \log\left[\int\mathbb{E}_{\boldsymbol{\xi}}\exp\left(\frac{\gamma}{n}\sum_{i=1}^n\xi_i uh(\boldsymbol{\theta},\mathbf{x}_i)\right)p_0(\boldsymbol{\theta},u)d\boldsymbol{\theta}du\right]\right\},
\end{aligned}
$$

where the first inequality follows by the Donsker-Varadhan representation of KL-divergence [14], and the second inequality follows by Jensen's inequality. Note that $\xi_1, \ldots, \xi_n$ are i.i.d. Rademacher random variables. By standard tail bound we have

$$
\mathbb{E}_{\boldsymbol{\xi}}\exp\left[\frac{\gamma}{n}\sum_{i=1}^n\xi_i uh(\boldsymbol{\theta},\mathbf{x}_i)\right] \leq \exp\left[\frac{\gamma^2}{2n^2}\sum_{i=1}^n u^2 h^2(\boldsymbol{\theta},\mathbf{x}_i)\right].
$$

Therefore

$$\mathfrak{R}_n(\mathcal{F}_{\mathrm{KL}}(M)) \leq \frac{\alpha}{\gamma} \cdot \left\{ M + \log \left[ \int \exp \left( \frac{\gamma^2}{2n^2} \sum_{i=1}^n u^2 h^2(\boldsymbol{\theta}, \mathbf{x}_i) \right) p_0(\boldsymbol{\theta}, u) d\boldsymbol{\theta} du \right] \right\}.$$

Now by the assumption that $h(\boldsymbol{\theta}, \mathbf{x}) \leq G$, we have

$$\int \exp \left( \frac{\gamma^2}{2n^2} \sum_{i=1}^n u^2 h^2(\boldsymbol{\theta}, \mathbf{x}_i) \right) p_0(\boldsymbol{\theta}, u) d\boldsymbol{\theta} du \leq \int \exp \left( \frac{\gamma^2 G^2}{2n} u^2 \right) p_0(\boldsymbol{\theta}, u) d\boldsymbol{\theta} du$$

$$= \frac{1}{\sqrt{2\pi}} \cdot \sqrt{\frac{2\pi}{1 - \gamma^2 G^2 n^{-1}}}$$

$$= \sqrt{\frac{1}{1 - \gamma^2 G^2 n^{-1}}}.$$

Therefore we have

$$\mathfrak{R}_n(\mathcal{F}_{\mathrm{KL}}(M))) \leq \frac{\alpha}{\gamma} \cdot \left[ M + \log \left( \sqrt{\frac{1}{1 - \gamma^2 G^2 n^{-1}}} \right) \right].$$

Setting $\gamma = G^{-1}\sqrt{Mn}$ and applying the inequality $\log(1 - z) \geq -2z$ for $z \in [0, 1/2]$ gives

$$\mathfrak{R}_n(\mathcal{F}_{\mathrm{KL}}(M))) \leq \frac{G\alpha}{\sqrt{Mn}} \cdot \left[ M + \log \left( \sqrt{\frac{1}{1 - M}} \right) \right] \leq 2G\alpha \sqrt{\frac{M}{n}}.$$

This completes the proof. $\qquad \square$

## A.5 Proof of Lemma 5.6

*Proof of Lemma 5.6.* We first introduce the following ramp loss function, which is frequently used in the analysis of generalization bounds [7, 23] for binary classification problems.

$$\ell_{\mathrm{ramp}}(y', y) = \begin{cases} 0 & \text{if } y'y \geq 1/2, \\ -2y'y + 1, & \text{if } 0 \leq y'y < 1/2, \\ 1, & \text{if } y'y < 0. \end{cases}$$

Then by definition, we see that $\ell_{\mathrm{ramp}}(y', y)$ is 2-Lipschitz in the first argument, $\ell_{\mathrm{ramp}}(y, y) = 0$, $|\ell_{\mathrm{ramp}}(y', y)| \leq 1$, and

$$\ell^{0\text{-}1}(y', y) \leq \ell_{\mathrm{ramp}}(y', y) \leq |y' - y| \qquad (A.25)$$

for all $y' \in \mathbb{R}$ and $y \in \{\pm 1\}$. By the Lipschitz and boundedness properties of the ramp loss, we apply the standard properties of Rademacher complexity [8, 30, 33] and obtain that with probability at least $1 - \delta$,

$$\mathbb{E}_{\mathcal{D}}[\ell_{\mathrm{ramp}}(f(\mathbf{x}), y)/2] \leq \mathbb{E}_S[\ell_{\mathrm{ramp}}(f(\mathbf{x}), y)/2] + 2\mathfrak{R}_n(\mathcal{F}_{\mathrm{KL}}(M)) + 3\sqrt{\frac{\log(2/\delta)}{2n}}$$

for all $f \in \mathcal{F}_{\mathrm{KL}}(M)$. Now we have

$$\mathbb{E}_{\mathcal{D}}[\ell^{0\text{-}1}(f(\mathbf{x}), y)] \leq 2\mathbb{E}_{\mathcal{D}}[\ell_{\mathrm{ramp}}(f(\mathbf{x}), y)/2]$$

$$\leq \mathbb{E}_S[\ell_{\mathrm{ramp}}(f(\mathbf{x}), y)] + 4\mathfrak{R}_n(\mathcal{F}_{\mathrm{KL}}(M)) + 6\sqrt{\frac{\log(2/\delta)}{2n}}$$

$$\leq \mathbb{E}_S[|f(\mathbf{x}) - y|] + 4\mathfrak{R}_n(\mathcal{F}_{\mathrm{KL}}(M)) + 6\sqrt{\frac{\log(2/\delta)}{2n}}$$

$$\leq \sqrt{\mathbb{E}_S[|f(\mathbf{x}) - y|^2]} + 4\mathfrak{R}_n(\mathcal{F}_{\mathrm{KL}}(M)) + 6\sqrt{\frac{\log(2/\delta)}{2n}}.$$

Here the first and third inequalities follow by the first and second parts of the inequality in (A.25) respectively, and the last inequality uses Jensen's inequality. This completes the proof. $\qquad \square$

# B Proof of Lemmas in Appendix A

In this section we provide the proof of technical lemmas we use in Appendix A.

## B.1 Proof of Lemma A.1

Here we provide the proof of Lemma A.1, which is essentially based on direct calculations on the activation function and the assumption that $\|\mathbf{x}\|_2 \leq 1$.

*Proof of Lemma A.1.* By $h(\boldsymbol{\theta}, \mathbf{x}) = \widetilde{h}(\boldsymbol{\theta}^\top \mathbf{x})$, we have the following identities.

$$\nabla_{\boldsymbol{\theta}} h(\boldsymbol{\theta}, \mathbf{x}) = \widetilde{h}'(\boldsymbol{\theta}^\top \mathbf{x})\mathbf{x}, \ \Delta h(\boldsymbol{\theta}^\top \mathbf{x}) = \sum_{i=1} \widetilde{h}''(\boldsymbol{\theta}^\top \mathbf{x})x_i^2 = \widetilde{h}''(\boldsymbol{\theta}^\top \mathbf{x})\|\mathbf{x}\|_2^2, \ \nabla_{\boldsymbol{\theta}} h(\boldsymbol{\theta}, \mathbf{x}) \cdot \boldsymbol{\theta} = \widetilde{h}'(\boldsymbol{\theta}^\top \mathbf{x})\boldsymbol{\theta}^\top \mathbf{x}.$$

By $|\widetilde{h}(z)| \leq G$ in Assumption 4.2 and $\|\mathbf{x}\|_2 \leq 1$ in Assumption 4.1, we have

$$|h(\boldsymbol{\theta}, \mathbf{x})| \leq G,$$

which gives the first bound. The other results can be derived similarly, which we present as follows. By $|\widetilde{h}'(z)| \leq G$ and $\|\mathbf{x}\|_2 \leq 1$, we have

$$\|\nabla_{\boldsymbol{\theta}} h(\boldsymbol{\theta}, \mathbf{x})\|_2 = \|\widetilde{h}'(\boldsymbol{\theta}^\top \mathbf{x})\mathbf{x}\|_2 \leq G,$$

which gives the second bound. By $|\widetilde{h}''(z)| \leq G$ and $\|\mathbf{x}\|_2 \leq 1$, we have

$$|\Delta h(\boldsymbol{\theta}, \mathbf{x})| = |\widetilde{h}''(\boldsymbol{\theta}^\top \mathbf{x})\|\mathbf{x}\|_2^2| \leq G.$$

Moreover, based on the same assumptions we also have

$$\begin{aligned}
\|\nabla_{\boldsymbol{\theta}} h(\boldsymbol{\theta}_1, \mathbf{x}) - \nabla_{\boldsymbol{\theta}} h(\boldsymbol{\theta}_2, \mathbf{x})\|_2 &= \|\widetilde{h}'(\boldsymbol{\theta}_1^\top \mathbf{x})\mathbf{x} - \widetilde{h}'(\boldsymbol{\theta}_2^\top \mathbf{x})\mathbf{x}\|_2 \\
&\leq |\widetilde{h}'(\boldsymbol{\theta}_1^\top \mathbf{x}) - \widetilde{h}'(\boldsymbol{\theta}_2^\top \mathbf{x})| \\
&\leq G|\boldsymbol{\theta}_1^\top \mathbf{x} - \boldsymbol{\theta}_2^\top \mathbf{x}| \\
&\leq G\|\boldsymbol{\theta}_1^\top - \boldsymbol{\theta}_2^\top\|_2.
\end{aligned}$$

Therefore the third and fourth bounds hold. Applying the bound $\left|\left(z\widetilde{h}'(z)\right)'\right| \leq G$ and $\|\mathbf{x}\|_2 \leq 1$ gives the fifth bound:

$$\|\nabla_{\boldsymbol{\theta}}\left(\nabla_{\boldsymbol{\theta}} h(\theta, \mathbf{x}) \cdot \boldsymbol{\theta}\right)\|_2 = \|\nabla_{\boldsymbol{\theta}}\left(\widetilde{h}'(\boldsymbol{\theta}^\top \mathbf{x})\boldsymbol{\theta}^\top \mathbf{x}\right)\|_2 = \|\mathbf{x}\|_2 \left|\left(z\widetilde{h}'(z)\right)'|_{z=\boldsymbol{\theta}^\top \mathbf{x}}\right| \leq G.$$

Finally, by $|\widetilde{h}'''(z)| \leq G$ and $\|\mathbf{x}\|_2 \leq 1$, we have

$$\|\nabla_{\boldsymbol{\theta}}\Delta_{\boldsymbol{\theta}} h(\boldsymbol{\theta}, \mathbf{x})\|_2 = \|\nabla_{\boldsymbol{\theta}}\widetilde{h}''(\boldsymbol{\theta}^\top \mathbf{x})\|_2 \|\mathbf{x}\|_2^2 \leq |\widetilde{h}'''(\boldsymbol{\theta}^\top \mathbf{x})|\|\mathbf{x}\|_2^3 \leq G.$$

This completes the proof. $\qquad\square$

## B.2 Proof of Lemma A.2

The following lemma bounds the second moment of a distribution $p$ that is close to $p_0$ in 2-Wasserstein distance.

**Lemma B.1.** For $\mathcal{W}_2(p, p_0) \leq \sqrt{d+1}$, the following bound holds:

$$\mathbb{E}_p(\|\boldsymbol{\theta}\|_2^2 + u^2) \leq 4d + 4$$

The following lemma is a reformulation of Lemma C.8 in Xu et al. [38]. For completeness, we provide its proof in Appendix B.

**Lemma B.2.** For $\mathcal{W}_2(p, p_0) \leq \sqrt{d+1}$, let $g(u, \boldsymbol{\theta}) : \mathbb{R}^{d+1} \to \mathbb{R}$ be a $C^1$ function such that

$$\sqrt{\nabla_u g(u, \boldsymbol{\theta})^2 + \|\nabla_{\boldsymbol{\theta}} g(u, \boldsymbol{\theta})\|^2} \leq C_1\sqrt{u^2 + \|\boldsymbol{\theta}\|_2^2} + C_2, \forall \mathbf{x} \in \mathbb{R}^{d'}$$

for some constants $C_1, C_2 \geq 0$. Then

$$\left|\mathbb{E}_p[g(u, \boldsymbol{\theta})] - \mathbb{E}_{p_0}[g(u_0, \boldsymbol{\theta}_0)]\right| \leq \left(2C_1\sqrt{d+1} + C_2\right)\mathcal{W}_2(p, p_0).$$

*Proof of Lemma A.2.* Let $\pi^*$ be the optimal coupling of $\mathcal{W}_2(p, p_0)$. Then we have

$$
\begin{aligned}
\big|\mathbf{H}_1(p)_{i,j} - \mathbf{H}_1(p_0)_{i,j}\big| &= \big|\mathbb{E}_{\pi^*}[u^2 \nabla_{\boldsymbol{\theta}} h(\boldsymbol{\theta}, \mathbf{x}_i) \cdot \nabla_{\boldsymbol{\theta}} h(\boldsymbol{\theta}, \mathbf{x}_j)] - \mathbb{E}_{\pi^*}[u_0^2 \nabla_{\boldsymbol{\theta}} h(\boldsymbol{\theta}_0, \mathbf{x}_i) \cdot \nabla_{\boldsymbol{\theta}} h(\boldsymbol{\theta}_0, \mathbf{x}_j)]\big| \\
&\leq \underbrace{\big|\mathbb{E}_{\pi^*}[(u^2 - u_0^2) \nabla_{\boldsymbol{\theta}} h(\boldsymbol{\theta}, x_i) \cdot \nabla_{\boldsymbol{\theta}} h(\boldsymbol{\theta}, x_j)]\big|}_{I_1} \\
&\quad + \underbrace{\big|\mathbb{E}_{\pi^*}\big[u_0^2\big(\nabla_{\boldsymbol{\theta}} h(\boldsymbol{\theta}, x_i) \cdot \nabla_{\boldsymbol{\theta}} h(\boldsymbol{\theta}, x_j) - \nabla_{\boldsymbol{\theta}} h(\boldsymbol{\theta}_0, x_i) \cdot \nabla_{\boldsymbol{\theta}} h(\boldsymbol{\theta}_0, x_j)\big)\big]\big|}_{I_2}.
\end{aligned}
$$
(B.1)

We first bound $I_1$ as follows.

$$
\begin{aligned}
I_1 &\leq G^2 \mathbb{E}_{\pi^*}[|u^2 - u_0^2|] \\
&\leq G^2 \sqrt{\mathbb{E}_{\pi^*}[(u - u_0)^2]} \sqrt{\mathbb{E}_{\pi^*}[(u + u_0)^2]} \\
&\leq G^2 \mathcal{W}_2(p, p_0) \sqrt{2\mathbb{E}_p[u^2] + 2\mathbb{E}_{p_0}[u_0^2]} \\
&\leq G^2 \mathcal{W}_2(p, p_0) \sqrt{8d + 10},
\end{aligned}
$$
(B.2)

where the first inequality is by $\|\nabla_{\boldsymbol{\theta}} h(\boldsymbol{\theta}, x_i)\|_2 \leq G$ in Lemma A.1, the second inequality is by Cauchy-Schwarz inequality, the third inequality is by Jensen's inequality and the last inequality is by Lemma B.1. Next, We bound $I_2$ in (B.1). For any given $r > 0$ we have

$$
\begin{aligned}
I_2 &\leq \mathbb{E}_{\pi^*}\big[u_0^2 \mathbb{1}(|u_0 < r|)\big|\nabla_{\boldsymbol{\theta}} h(\boldsymbol{\theta}, x_i) \cdot \nabla_{\boldsymbol{\theta}} h(\boldsymbol{\theta}, x_j) - \nabla_{\boldsymbol{\theta}} h(\boldsymbol{\theta}_0, x_i) \cdot \nabla_{\boldsymbol{\theta}} h(\boldsymbol{\theta}_0, x_j)\big|\big] \\
&\quad + \mathbb{E}_{\pi^*}\big[u_0^2 \mathbb{1}(|u_0 \geq r|)\big|\nabla_{\boldsymbol{\theta}} h(\boldsymbol{\theta}, x_i) \cdot \nabla_{\boldsymbol{\theta}} h(\boldsymbol{\theta}, x_j) - \nabla_{\boldsymbol{\theta}} h(\boldsymbol{\theta}_0, x_i) \cdot \nabla_{\boldsymbol{\theta}} h(\boldsymbol{\theta}_0, x_j)\big|\big] \\
&\leq r^2 \mathbb{E}_{\pi^*}\big[\big|\nabla_{\boldsymbol{\theta}} h(\boldsymbol{\theta}, x_i) \cdot \nabla_{\boldsymbol{\theta}} h(\boldsymbol{\theta}, x_j) - \nabla_{\boldsymbol{\theta}} h(\boldsymbol{\theta}_0, x_i) \cdot \nabla_{\boldsymbol{\theta}} h(\boldsymbol{\theta}_0, x_j)\big|\big] + 2G^2 \mathbb{E}_{\pi^*}[u_0^2 \mathbb{1}(|u_0 \geq r|)],
\end{aligned}
$$
(B.3)

where the second inequality is by $\|\nabla_{\boldsymbol{\theta}} h(\boldsymbol{\theta}, x_i)\|_2 \leq G$ Lemma A.1. We further bound the first term on the right-hand side of (B.3),

$$
\begin{aligned}
\mathbb{E}_{\pi^*}\big[\big|\nabla_{\boldsymbol{\theta}} h(\boldsymbol{\theta}, x_i) \cdot \nabla_{\boldsymbol{\theta}} h(\boldsymbol{\theta}, x_j) &- \nabla_{\boldsymbol{\theta}} h(\boldsymbol{\theta}_0, x_i) \cdot \nabla_{\boldsymbol{\theta}} h(\boldsymbol{\theta}_0, x_j)\big|\big] \\
&\leq \mathbb{E}_{\pi^*}\big[\big|\nabla_{\boldsymbol{\theta}} h(\boldsymbol{\theta}, x_i) \cdot \big(\nabla_{\boldsymbol{\theta}} h(\boldsymbol{\theta}, x_j) - \nabla_{\boldsymbol{\theta}} h(\boldsymbol{\theta}_0, x_j)\big)\big|\big] \\
&\quad + \mathbb{E}_{\pi^*}\big[\big|\nabla_{\boldsymbol{\theta}} h(\boldsymbol{\theta}_0, x_j) \cdot \big(\nabla_{\boldsymbol{\theta}} h(\boldsymbol{\theta}, x_i) - \nabla_{\boldsymbol{\theta}} h(\boldsymbol{\theta}_0, x_i)\big)\big|\big] \\
&\leq 2G^2 \mathcal{W}_2(p, p_0),
\end{aligned}
$$
(B.4)

where the last inequality is by $\|\nabla_{\boldsymbol{\theta}} h(\boldsymbol{\theta}, \mathbf{x})\|_2 \leq G$ and $\|\nabla_{\boldsymbol{\theta}} h(\boldsymbol{\theta}, \mathbf{x}) - \nabla_{\boldsymbol{\theta}} h(\boldsymbol{\theta}_0, \mathbf{x})\|_2 \leq G\|\boldsymbol{\theta} - \boldsymbol{\theta}_0\|_2$ in Lemma A.1. Plugging (B.4) into (B.3) yields

$$
I_2 \leq 2r^2 G^2 \mathcal{W}_2(p, p_0) + 2G^2 \mathbb{E}_{\pi^*}[u_0^2 \mathbb{1}(|u_0 \geq r|)].
$$
(B.5)

Further plugging (B.2) and (B.5) into (B.1), we obtain

$$
\begin{aligned}
\big|\mathbf{H}_1(p)_{i,j} - \mathbf{H}_1(p_0)_{i,j}\big| &\leq G^2 \mathcal{W}_2(p, p_0) \sqrt{8d + 10} + 2r^2 G^2 \mathcal{W}_2(p, p_0) \\
&\quad + 2G^2 \mathbb{E}_{p_0}[u_0^2 \mathbb{1}(|u_0 \geq r|)].
\end{aligned}
$$

This finishes the proof. $\qquad\square$

## B.3 Proof of Lemma A.3

Here we provide the proof of Lemma A.3, which is essentially based on a direct application of Lemma A.1 and the definition of 2-Wasserstein distance.

*Proof of Lemma A.3.* Denote $\widehat{H}_{i,j}(\boldsymbol{\theta}, u) = h(\boldsymbol{\theta}, \mathbf{x}_i) h(\boldsymbol{\theta}, \mathbf{x}_j)$, then we have $\mathbf{H}_2(p)_{i,j} = \mathbb{E}_p[\widehat{H}_{i,j}(\boldsymbol{\theta}, u)]$. Calculating the gradient of $\widehat{H}_{i,j}(\boldsymbol{\theta}, u)$, we have

$$
\nabla_u \widehat{H}_{i,j}(\boldsymbol{\theta}, u) = 0, \qquad \|\nabla_{\boldsymbol{\theta}} \widehat{H}_{i,j}(\boldsymbol{\theta}, u)\|_2 \leq 2\|\nabla_{\boldsymbol{\theta}} h(\boldsymbol{\theta}, \mathbf{x}_i)\|_2 |h(\boldsymbol{\theta}, \mathbf{x}_j)| \leq 2G^2,
$$

where the second inequality is by Lemma A.1 . Applying Lemma B.2 gives

$$
|\mathbf{H}_2(p)_{i,j} - \mathbf{H}_2(p_0)_{i,j}| \leq 2G^2 \mathcal{W}_2(p, p_0).
$$

This finializes our proof. $\qquad\square$

## B.4 Proof of Lemma A.4

Lemma A.4 gives a tail bound on $p_0$, which is essentially a basic property of Gaussian distribution. For completeness we present the detailed proof as follows.

*Proof of Lemma A.4.* By the definition of $p_0$ we have

$$\mathbb{E}_{p_0}[u_0^2 \, \mathbb{1}(|u_0| \geq r)] = \frac{2}{\sqrt{2\pi}} \int_r^\infty u_0^2 \exp(-u_0^2/2) du_0 = \frac{2}{\sqrt{\pi}} \int_{r^2/2}^\infty t^{1/2} \exp(-t) dt$$

Now by the fact that $4z/\pi \leq \exp(z), \forall z \in \mathbb{R}$, we have

$$\mathbb{E}_{p_0}[u_0^2 \, \mathbb{1}(|u_0 \geq r|)] \leq \int_{r^2/2}^\infty \exp(-t/2) dt = \frac{1}{2} \exp\left(-\frac{r^2}{4}\right),$$

which finalizes our proof. $\qquad\square$

## B.5 Proof of Lemma A.5

We first introduce some notations on the first variations. For $i \in [n]$, $\frac{\partial \mathbf{f}(t)_i}{\partial p_t}$, $\frac{\partial L(p_t)}{\partial p_t}$, $\frac{\partial D_{\mathrm{KL}}(p_t||p_0)}{\partial p_t}$ and $\frac{\partial Q(p_t)}{\partial p_t}$ are defined as follows.

$$\frac{\partial \mathbf{f}(t)_i}{\partial p_t} := \alpha u h(\boldsymbol{\theta}, \mathbf{x}_i), \tag{B.6}$$

$$\frac{\partial L(p_t)}{\partial p_t} := \mathbb{E}_S\big[\nabla_{y'}\phi\big(f(p_t, \mathbf{x}), y\big) \cdot \alpha u h(\boldsymbol{\theta}, \mathbf{x})\big], \tag{B.7}$$

$$\frac{\partial D_{\mathrm{KL}}(p_t||p_0)}{\partial p_t} := \log(p_t/p_0) + 1, \tag{B.8}$$

$$\frac{\partial Q(p_t)}{\partial p_t} := \frac{\partial L(p_t)}{\partial p_t} + \lambda \frac{\partial D_{\mathrm{KL}}(p_t||p_0)}{\partial p_t}$$
$$= \mathbb{E}_S\big[\nabla_{y'}\phi\big(f(p_t, \mathbf{x}), y\big) \cdot \alpha u h(\boldsymbol{\theta}, \mathbf{x}) + \lambda \log(p_t/p_0) + \lambda\big]. \tag{B.9}$$

The following lemma summarizes some direct calculations on the relation between these first variations defined above and the time derivatives of $\mathbf{f}(t)_i$, $L(p_t)$, $D_{\mathrm{KL}}(p_t||p_0)$ and $Q(p_t)$. Note that these results are well-known results in literature, but for completeness we present the detailed calculations in Appendix C.3.

**Lemma B.3.** Let $\frac{\partial \mathbf{f}(t)_i}{\partial p_t}$, $\frac{\partial L(p_t)}{\partial p_t}$, $\frac{\partial D_{\mathrm{KL}}(p_t||p_0)}{\partial p_t}$, $\frac{\partial Q(p_t)}{\partial p_t}$ be the first variations defined in (B.6), (B.7), (B.8) and (B.9). Then

$$\frac{\partial[\mathbf{f}(t)_i - y_i]}{\partial t} = \int_{\mathbb{R}^{d+1}} \frac{\partial \mathbf{f}(t)_i}{\partial p_t} \frac{dp_t}{dt} d\boldsymbol{\theta} du,$$

$$\frac{\partial L(p_t)}{\partial t} = \int_{\mathbb{R}^{d+1}} \frac{\partial L(p_t)}{\partial p_t} \frac{dp_t}{dt} d\boldsymbol{\theta} du,$$

$$\frac{\partial D_{\mathrm{KL}}(p_t||p_0)}{\partial t} = \int_{\mathbb{R}^{d+1}} \frac{\partial D_{\mathrm{KL}}(p_t||p_0)}{\partial p_t} \frac{dp_t}{dt} d\boldsymbol{\theta} du,$$

$$\frac{\partial Q(p_t)}{\partial t} = \int_{\mathbb{R}^{d+1}} \frac{\partial Q(p_t)}{\partial p_t} \frac{dp_t}{dt} d\boldsymbol{\theta} du.$$

The following lemma summarizes the calculation of the gradients of the first variations defined in (B.7), (B.8) and (B.9).

**Lemma B.4.** Let $\frac{\partial L(p_t)}{\partial p_t}$, $\frac{\partial D_{\mathrm{KL}}(p_t\|p_0)}{\partial p_t}$ and $\frac{\partial Q(p_t)}{\partial p_t}$ be the first variations defined in (B.7), (B.8) and (B.9). Then their gradients with respect to $u$ and $\boldsymbol{\theta}$ are given as follows:

$$\nabla_u \frac{\partial L(p_t)}{\partial p_t} = -\widehat{g}_1(t, \boldsymbol{\theta}, u), \nabla_{\boldsymbol{\theta}} \frac{\partial L(p_t)}{\partial p_t} = -\widehat{g}_2(t, \boldsymbol{\theta}, u),$$

$$\nabla_u \frac{\partial D_{\mathrm{KL}}(p_t\|p_0)}{\partial p_t} = u + \nabla_{\boldsymbol{\theta}} \log(p_t), \nabla_{\boldsymbol{\theta}} \frac{\partial D_{\mathrm{KL}}(p_t\|p_0)}{\partial p_t} = \boldsymbol{\theta} + \nabla_{\boldsymbol{\theta}} \log(p_t),$$

$$\nabla \frac{\partial Q(p_t)}{\partial p_t} = \nabla \frac{\partial L(p_t)}{\partial p_t} + \lambda \nabla \frac{\partial D_{\mathrm{KL}}(p_t\|p_0)}{\partial p_t}.$$

Moreover, the PDE (3.4) can be written as

$$\frac{dp_t}{dt} = \nabla \cdot \left[ p_t(\boldsymbol{\theta}, u) \nabla \frac{\partial Q(p_t)}{\partial p_t} \right].$$

*Proof of Lemma A.5.* By Lemma B.3, we have the following chain rule

$$\begin{aligned}
\frac{\partial L(p_t)}{\partial t} &= \int_{\mathbb{R}^{d+1}} \frac{\partial L(p_t)}{\partial p_t} \frac{dp_t}{dt} d\boldsymbol{\theta} du \\
&= \int_{\mathbb{R}^{d+1}} \frac{\partial L(p_t)}{\partial p_t} \nabla \cdot \left[ p_t(\boldsymbol{\theta}, u) \nabla \frac{\partial Q(p_t)}{\partial p_t} \right] d\boldsymbol{\theta} du \\
&= -\int_{\mathbb{R}^{d+1}} p_t(\boldsymbol{\theta}, u) \left[ \nabla \frac{\partial L(p_t)}{\partial p_t} \right] \cdot \left[ \nabla \frac{\partial Q(p_t)}{\partial p_t} \right] d\boldsymbol{\theta} du \\
&= \underbrace{-\int_{\mathbb{R}^{d+1}} p_t(\boldsymbol{\theta}, u) \left[ \nabla \frac{\partial L(p_t)}{\partial p_t} \right] \cdot \left[ \nabla \frac{\partial L(p_t)}{\partial p_t} \right] d\boldsymbol{\theta} du}_{I_1} \\
&\quad \underbrace{- \lambda \int_{\mathbb{R}^{d+1}} p_t(\boldsymbol{\theta}, u) \left[ \nabla \frac{\partial L(p_t)}{\partial p_t} \right] \cdot \left[ \nabla \frac{\partial D(p_t\|p_0)}{\partial p_t} \right] d\boldsymbol{\theta} du}_{I_2}, \quad (\text{B.10})
\end{aligned}$$

where the second and last equation is by Lemma B.4, the third inequality is by apply integration by parts. We now proceed to calculate $I_1$ and $I_2$ based on the calculations of derivatives in Lemma B.4. For $I_1$, we have

$$I_1 = \int_{\mathbb{R}^{d+1}} p_t(\boldsymbol{\theta}, u) \|\widehat{g}_1(t, \boldsymbol{\theta}, u)\|_2^2 d\boldsymbol{\theta} du + \int_{\mathbb{R}^{d+1}} p_t(\boldsymbol{\theta}, u) |\widehat{g}_2(t, \boldsymbol{\theta}, u)|^2 d\boldsymbol{\theta} d\boldsymbol{\theta} du. \quad (\text{B.11})$$

Similarly, for $I_2$, we have

$$\begin{aligned}
I_2 &= \int_{\mathbb{R}^{d+1}} p_t(\boldsymbol{\theta}, u) [-\widehat{g}_1(t, \boldsymbol{\theta}, u)] \cdot [u + \nabla_u \log(p_t)] d\boldsymbol{\theta} du \\
&\quad + \int_{\mathbb{R}^{d+1}} p_t(\boldsymbol{\theta}, u) [-\widehat{g}_2(t, \boldsymbol{\theta}, u)] \cdot [\boldsymbol{\theta} + \nabla_{\boldsymbol{\theta}} \log(p_t)] d\boldsymbol{\theta} du \\
&= -\int_{\mathbb{R}^{d+1}} p_t(\boldsymbol{\theta}, u) [\widehat{g}_1(t, \boldsymbol{\theta}, u) \cdot u + \widehat{g}_2(t, \boldsymbol{\theta}, u)\boldsymbol{\theta}] d\boldsymbol{\theta} du \\
&\quad - \int_{\mathbb{R}^{d+1}} [\widehat{g}_1(t, \boldsymbol{\theta}, u) \cdot \nabla_u p_t(t, \boldsymbol{\theta}, u) + \widehat{g}_2(t, \boldsymbol{\theta}, u) \cdot \nabla_{\boldsymbol{\theta}} p_t(t, \boldsymbol{\theta}, u)] d\boldsymbol{\theta} du \\
&= -\int_{\mathbb{R}^{d+1}} p_t(\boldsymbol{\theta}, u) [\widehat{g}_1(t, \boldsymbol{\theta}, u) \cdot u + \widehat{g}_2(t, \boldsymbol{\theta}, u)\boldsymbol{\theta}] d\boldsymbol{\theta} du \\
&\quad + \int_{\mathbb{R}^{d+1}} p_t(t, \boldsymbol{\theta}, u) [\nabla_u \cdot \widehat{g}_1(t, \boldsymbol{\theta}, u) + \nabla_{\boldsymbol{\theta}} \cdot \widehat{g}_2(t, \boldsymbol{\theta}, u)] d\boldsymbol{\theta} du, \quad (\text{B.12})
\end{aligned}$$

where the second equation is by $p_t \nabla \log(p_t) = \nabla p_t$ and the third equation is by applying integration by parts. Plugging (B.11) and (B.12) into (B.10), we get

$$\begin{aligned}
\frac{\partial L(p_t)}{\partial t} &= -\int_{\mathbb{R}^{d+1}} p_t(\boldsymbol{\theta}, u) \|\widehat{g}_1(t, \boldsymbol{\theta}, u)\|_2^2 d\boldsymbol{\theta} du - \int_{\mathbb{R}^{d+1}} p_t(\boldsymbol{\theta}, u) |\widehat{g}_2(t, \boldsymbol{\theta}, u)|^2 d\boldsymbol{\theta} d\boldsymbol{\theta} du \\
&\quad + \lambda \int_{\mathbb{R}^{d+1}} p_t(\boldsymbol{\theta}, u) [\widehat{g}_1 \cdot u + \widehat{g}_2 \cdot \boldsymbol{\theta} - \nabla_u \cdot \widehat{g}_1 - \nabla_{\boldsymbol{\theta}} \cdot \widehat{g}_2] d\boldsymbol{\theta} du.
\end{aligned}$$

This completes the proof. □

## B.6 Proof of Lemma A.6

Here we prove Lemma A.6, which is based on its connection to the Gram matrix of neural tangent kernel.

*Proof of Lemma A.6.* We first remind the readers of the definitions of the Gram matrices in (3.6). Let $\mathbf{b}(p_t) = (f(p_t, \mathbf{x}_1) - y_1, \ldots, f(p_t, \mathbf{x}_n) - y_n)^\top \in \mathbb{R}^n$. Then by the definitions of $\mathbf{H}_1(p_t)$ and $\mathbf{H}_2(p_t)$ in (3.6), we have

$$\int_{\mathbb{R}^{d+1}} p_t(\boldsymbol{\theta}, u) \big[ |\mathbb{E}_S[(f(p_t, \mathbf{x}) - y)h(\boldsymbol{\theta}, \mathbf{x})]|^2 d\boldsymbol{\theta} du = \frac{1}{n^2} \mathbf{b}(p_t)^\top \mathbf{H}_1(p_t)\mathbf{b}(p_t),$$

$$\int_{\mathbb{R}^{d+1}} p_t(\boldsymbol{\theta}, u) \big[ \|\mathbb{E}_S[(f(p_t, \mathbf{x}) - y)u\nabla_{\boldsymbol{\theta}} h(\boldsymbol{\theta}, \mathbf{x})]\|_2^2 \big] d\boldsymbol{\theta} du = \frac{1}{n^2} \mathbf{b}(p_t)^\top \mathbf{H}_2(p_t)\mathbf{b}(p_t).$$

Therefore by (3.6) we have

$$\int_{\mathbb{R}^{d+1}} p_t(\boldsymbol{\theta}, u) \big[ |\mathbb{E}_S[(f(p_t, \mathbf{x}) - y)h(\boldsymbol{\theta}, \mathbf{x})]|^2 + \|\mathbb{E}_S[(f(p_t, \mathbf{x}) - y)u\nabla_{\boldsymbol{\theta}} h(\boldsymbol{\theta}, \mathbf{x})]\|_2^2 \big] d\boldsymbol{\theta} du$$

$$= \frac{1}{n^2} \mathbf{b}(p_t)^\top \mathbf{H}(p_t)\mathbf{b}(p_t). \tag{B.13}$$

By the definition of $t^*$, for $t \leq t^*$ we have $\mathcal{W}_2(p_t, p_0) \leq R$, and therefore applying Lemma 5.1 gives

$$\frac{1}{n^2} \mathbf{b}(p_t)^\top \mathbf{H}(p_t)\mathbf{b}(p_t) \geq \frac{\Lambda \|\mathbf{b}(p_t)\|_2^2}{2n^2} = \frac{\lambda_0^2}{2} L(p_t), \tag{B.14}$$

where the equation follows by the definition of $\mathbf{b}(p_t)$. Plugging (B.14) into (B.13) completes the proof. □

## B.7 Proof of Lemma A.7

**Lemma B.5.** Under Assumptions 4.1 and 4.2, for all $\mathcal{W}(p, p_0) \leq \sqrt{d+1}$ and $\mathbf{x}$ the following inequality holds.

$$\big| \mathbb{E}_p \big[ uh(\boldsymbol{\theta}, \mathbf{x}) + u\nabla h(\boldsymbol{\theta}, \mathbf{x}) \cdot \boldsymbol{\theta} - u\Delta h(\boldsymbol{\theta}, \mathbf{x}) \big] \big| \leq A_1,$$

where $A_1$ is defined in Theorem 4.4.

The proof of Lemma A.7 is based on direct applications of Lemma B.5. We present the proof as follows.

*Proof of Lemma A.7.* We have the following identities:

$$\widehat{g}_1(t, \boldsymbol{\theta}, u) = -\mathbb{E}_S[\nabla_f \phi(f(p_t, \mathbf{x}), y)\alpha h(\boldsymbol{\theta}, \mathbf{x})],$$
$$\widehat{g}_2(t, \boldsymbol{\theta}, u) = -\mathbb{E}_S[\nabla_f \phi(f(p_t, \mathbf{x}), y)\alpha u\nabla_{\boldsymbol{\theta}} h(\boldsymbol{\theta}, \mathbf{x})],$$
$$\nabla_u \cdot \widehat{g}_1(t, \boldsymbol{\theta}, u) = 0,$$
$$\nabla_{\boldsymbol{\theta}} \cdot \widehat{g}_2(t, \boldsymbol{\theta}, u) = -\mathbb{E}_S[\nabla_f \phi(f(p_t, \mathbf{x}), y)\alpha u\Delta h(\boldsymbol{\theta}, \mathbf{x})].$$

Base on these identities we can derive

$$\left| \int_{\mathbb{R}^{d+1}} p_t(\boldsymbol{\theta}, u)[\widehat{g}_1 \cdot u + \widehat{g}_2 \cdot \boldsymbol{\theta} - \nabla_u \cdot \widehat{g}_1 - \nabla_{\boldsymbol{\theta}} \cdot \widehat{g}_2] d\boldsymbol{\theta} du \right|$$

$$= \left| \alpha \mathbb{E}_S \Big[ \nabla_f \phi(f(p_t, \mathbf{x}), y) \mathbb{E}_{p_t} \Big[ \big( u_t h(\boldsymbol{\theta}_t, \mathbf{x}_i) + u_t \nabla h(\boldsymbol{\theta}_t, \mathbf{x}_i) \cdot \boldsymbol{\theta}_t - u_t \Delta h(\boldsymbol{\theta}_t, \mathbf{x}_i) \big) \Big] \Big] \right|$$

$$\leq 2\alpha A_1 \mathbb{E}_S[|f(p_t, \mathbf{x}) - y|]$$

$$\leq 2\alpha A_1 \sqrt{L(p_t)},$$

where the first inequality is by Lemma B.5, the second inequality is by Jensen's inequality. □

## B.8 Proof of Lemma A.8

The following lemma summarizes the calculation on the time derivative of $D_{\mathrm{KL}}(p_t || p_0)$.

**Lemma B.6.** Let $p_t$ be the solution of PDE (3.4). Then the following identity holds.

$$\frac{\partial D_{\mathrm{KL}}(p_t||p_0)}{\partial t} = -\lambda \int_{\mathbb{R}^{d+1}} p_t(\boldsymbol{\theta}, u) \|\boldsymbol{\theta} + \nabla_{\boldsymbol{\theta}} \log(p_t)\|_2^2 - \lambda \int_{\mathbb{R}^{d+1}} p_t(\boldsymbol{\theta}, u) |u + \nabla_u \log(p_t)|^2$$
$$+ \int_{\mathbb{R}^{d+1}} p_t(\boldsymbol{\theta}, u) [\widehat{g}_1 \cdot u + \widehat{g}_2 \cdot \boldsymbol{\theta} - \nabla_u \cdot \widehat{g}_1 - \nabla_{\boldsymbol{\theta}} \cdot \widehat{g}_2] d\boldsymbol{\theta} du.$$

In the calculation given by Lemma B.6, we can see that the (potentially) positive term in $\frac{\partial D_{\mathrm{KL}}(p_t||p_0)}{\partial t}$ naturally coincides with the corresponding term in $\frac{\partial L(p_t)}{\partial t}$ given by Lemma A.5, and a bound of it has already been given in Lemma A.7. However, for the analysis of the KL-divergence term, we present the following new bound, which eventually leads to a sharper result.

**Lemma B.7.** Under Assumptions 4.1 and 4.2, let $A_2$ be defined in Theorem 4.4. Then for $t \leq t^*$, it holds that

$$\int_{\mathbb{R}^{d+1}} p(\boldsymbol{\theta}, u) [\widehat{g}_1 \cdot u + \widehat{g}_2 \cdot \boldsymbol{\theta} - \nabla_u \cdot \widehat{g}_1 - \nabla_{\boldsymbol{\theta}} \cdot \widehat{g}_2] d\boldsymbol{\theta} du \leq 2\alpha A_2 \sqrt{L(p_t)} \sqrt{D_{\mathrm{KL}}(p_t||p_0)}.$$

*Proof of Lemma A.8.* By Lemma B.6,

$$\frac{\partial D_{\mathrm{KL}}(p_t||p_0)}{\partial t} = -\lambda \int_{\mathbb{R}^{d+1}} p_t(\boldsymbol{\theta}, u) \|\boldsymbol{\theta} + \nabla_{\boldsymbol{\theta}} \log(p_t)\|_2^2 - \lambda \int_{\mathbb{R}^{d+1}} p_t(\boldsymbol{\theta}, u) |u + \nabla_u \log(p_t)|^2$$
$$+ \int_{\mathbb{R}^{d+1}} p_t(\boldsymbol{\theta}, u) [\widehat{g}_1 \cdot u + \widehat{g}_2 \cdot \boldsymbol{\theta} - \nabla_u \cdot \widehat{g}_1 - \nabla_{\boldsymbol{\theta}} \cdot \widehat{g}_2] d\boldsymbol{\theta} du$$
$$\leq 2A_2 \alpha \sqrt{L(p_t)} \sqrt{D_{\mathrm{KL}}(p_t||p_0)}, \tag{B.15}$$

where the inequality is by Lemma B.7. Notice that $\sqrt{D_{\mathrm{KL}}(p_0||p_0)} = 0$, $\sqrt{D_{\mathrm{KL}}(p_t||p_0)}$ is differentiable at $\sqrt{D_{\mathrm{KL}}(p_t||p_0)} \neq 0$ and from (B.15) the derivative

$$\frac{\partial \sqrt{D_{\mathrm{KL}}(p_t||p_0)}}{\partial t} = \frac{\partial D_{\mathrm{KL}}(p_t||p_0))}{\partial t} \frac{1}{2\sqrt{D_{\mathrm{KL}}(p_t||p_0)}} \leq A_2 \alpha \sqrt{L(p_t)},$$

which implies

$$\sqrt{D_{\mathrm{KL}}(p_t||p_0)} \leq \int_0^t A_2 \alpha \sqrt{L(p_s)} ds$$
$$\leq A_2 \alpha \int_0^t \exp(-\alpha^2 \lambda_0^2 s) + A_1 \lambda \alpha^{-1} \lambda_0^{-2} ds$$
$$\leq A_2 \alpha^{-1} \lambda_0^{-2} + A_2 A_1 \lambda \lambda_0^{-2} t,$$

where the second inequality holds due to Lemma 5.2. Squaring both sides and applying Jensen's inequality now gives

$$D_{\mathrm{KL}}(p_t||p_0) \leq 2A_2^2 \alpha^{-2} \lambda_0^{-4} + 2A_2^2 A_1^2 \lambda^2 \lambda_0^{-4} t^2.$$

This completes the proof. $\qquad \square$

## B.9 Proof of Lemma A.9

*Proof of Lemma A.9.* By Lemma B.3, we get

$$\frac{\partial Q(p_t)}{\partial t} = \int_{\mathbb{R}^{d+1}} \frac{\partial Q(p_t)}{\partial p_t} \frac{dp_t}{dt} d\boldsymbol{\theta} du$$
$$= \int_{\mathbb{R}^{d+1}} \frac{\partial Q(p_t)}{\partial p_t} \nabla \cdot \left[ p_t(\boldsymbol{\theta}, u) \nabla \frac{\partial Q(p_t)}{\partial p_t} \right] d\boldsymbol{\theta} du$$
$$= -\int_{\mathbb{R}^{d+1}} p_t(\boldsymbol{\theta}, u) \left\| \nabla \frac{\partial Q(p_t)}{\partial p_t} \right\|_2^2 d\boldsymbol{\theta} du$$
$$= -\int_{\mathbb{R}^{d+1}} p_t(\boldsymbol{\theta}, u) \|\widehat{g}_2 - \lambda\boldsymbol{\theta} - \lambda\nabla_{\boldsymbol{\theta}} \log(p_t)\|_2^2 - \int_{\mathbb{R}^{d+1}} p_t(\boldsymbol{\theta}, u) |\widehat{g}_1 - \lambda u - \lambda\nabla_u \log(p_t)|^2$$
$$\leq 0,$$

where the third equation is by applying integration by parts and the fourth equation is by Lemma B.4. □

# C  Proof of Auxiliary Lemmas in Appendix B

## C.1  Proof of Lemma B.1

*Proof of Lemma B.1.* Let $\pi^*(p_0, p)$ be the coupling that achieves the 2-Wasserstein distance between $p_0$ and $p$. Then by definition,

$$
\begin{aligned}
\mathbb{E}_{\pi^*}(\|\boldsymbol{\theta}\|_2^2 + u^2) &\leq \mathbb{E}_{\pi^*}(2\|\boldsymbol{\theta} - \boldsymbol{\theta}_0\|_2^2 + 2\|\boldsymbol{\theta}_0\|_2^2 + 2(u - u_0)^2 + 2u_0^2) \\
&\leq 2R^2 + 2d + 2 \\
&\leq 4d + 4,
\end{aligned}
$$

where the last inequality is by the assumption that $\mathcal{W}_2(p, p_0) \leq \sqrt{d+1}$. This finishes the proof. □

## C.2  Proof of Lemma B.2

*Proof of Lemma B.2.* By Lemma C.8 in Xu et al. [38], we have that

$$
\left| \mathbb{E}_p[g(u, \boldsymbol{\theta})] - \mathbb{E}_{p_0}[g(u_0, \boldsymbol{\theta}_0)] \right| \leq (C_1\sigma + C_2)\mathcal{W}_2(p, p_0),
$$

where $\sigma^2 = \max\{\mathbb{E}_p[u^2 + \boldsymbol{\theta}^2], \mathbb{E}_{p_0}[u_0^2 + \boldsymbol{\theta}_0^2]\}$. Then by Lemma B.1, we get $\sigma \leq 2\sqrt{d+1}$. Substituting the upper bound of $\sigma$ into the above inequality completes the proof. □

## C.3  Proof of Lemma B.3

*Proof of Lemma B.3.* By chain rule and the definition of $\mathbf{f}(t)$, we have

$$
\begin{aligned}
\frac{\partial[\mathbf{f}(t)_i - y_i]}{\partial t} &= \frac{d}{dt} \int_{\mathbb{R}^{d+1}} \alpha u h(\boldsymbol{\theta}, \mathbf{x}_i) p_t(\boldsymbol{\theta}, u) d\boldsymbol{\theta} du \\
&= \int_{\mathbb{R}^{d+1}} \alpha u h(\boldsymbol{\theta}, \mathbf{x}_i) \frac{dp_t}{dt}(\boldsymbol{\theta}, u) d\boldsymbol{\theta} du \\
&= \int_{\mathbb{R}^{d+1}} \frac{\partial \mathbf{f}(t)_i}{\partial p_t} \frac{dp_t}{dt} d\boldsymbol{\theta} du,
\end{aligned}
$$

where the last equation follows by the definition of the first variation $\frac{\partial \mathbf{f}(t)_i}{\partial p_t}$. This proves the first identity. Now we bound the second identity,

$$
\begin{aligned}
\frac{\partial L(p_t)}{\partial t} &= \mathbb{E}_S\left[\nabla_{y'}\phi\big(f(p_t, \mathbf{x}), y\big) \frac{d}{dt} f(p_t, \mathbf{x})\right] \\
&= \mathbb{E}_S\left[\nabla_{y'}\phi\big(f(p_t, \mathbf{x}), y\big) \frac{d}{dt} \int_{\mathbb{R}^{d+1}} \alpha u h(\boldsymbol{\theta}, \mathbf{x}) p_t(\boldsymbol{\theta}, u) d\boldsymbol{\theta} du\right] \\
&= \mathbb{E}_S\left[\nabla_{y'}\phi\big(f(p_t, \mathbf{x}), y\big) \int_{\mathbb{R}^{d+1}} \alpha u h(\boldsymbol{\theta}, \mathbf{x}) \frac{dp_t(\boldsymbol{\theta}, u)}{dt} d\boldsymbol{\theta} du\right] \\
&= \int_{\mathbb{R}^{d+1}} \frac{\partial L(p_t)}{\partial p_t} \frac{dp_t}{dt} d\boldsymbol{\theta} du,
\end{aligned}
$$

where the last equation follows by the definition of the first variation $\frac{\partial L(p_t)}{\partial p_t}$. This proves the second identity. Similarly, for $\frac{\partial D_{\mathrm{KL}}(p_t \| p_0)}{\partial t}$, we have

$$
\frac{\partial D_{\mathrm{KL}}(p_t \| p_0)}{\partial t} = \frac{d}{dt} \int p_t \log(p_t/p_0) d\boldsymbol{\theta} du = \int \frac{dp_t}{dt} \log(p_t/p_0) + \frac{dp_t}{dt} d\boldsymbol{\theta} du = \int_{\mathbb{R}^{d+1}} \frac{\partial D_{\mathrm{KL}}(p_t \| p_0)}{\partial p_t} \frac{dp_t}{dt} d\boldsymbol{\theta} du.
$$

Notice that $Q(p_t) = L(p_t) + \lambda D_{\mathrm{KL}}(p_t||p_0)$, so we have

$$\frac{\partial Q(p_t)}{\partial t} = \frac{\partial L(p_t)}{\partial t} + \lambda \frac{\partial D_{\mathrm{KL}}(p_t||p_0)}{\partial t}$$

$$= \int_{\mathbb{R}^{d+1}} \frac{\partial L(p_t)}{\partial p_t} \frac{dp_t}{dt} d\boldsymbol{\theta} du + \lambda \int_{\mathbb{R}^{d+1}} \frac{\partial D_{\mathrm{KL}}(p_t||p_0)}{\partial p_t} \frac{dp_t}{dt} d\boldsymbol{\theta} du$$

$$= \int_{\mathbb{R}^{d+1}} \frac{\partial Q(p_t)}{\partial p_t} \frac{dp_t}{dt} d\boldsymbol{\theta} du,$$

where the last equation is by the definition $\frac{\partial Q(p_t)}{\partial p_t} = \frac{\partial L(p_t)}{\partial p_t} + \lambda \frac{\partial D_{\mathrm{KL}}(p_t||p_0)}{\partial pt}$. This completes the proof. $\qquad \square$

### C.4 Proof of Lemma B.4

*Proof of Lemma B.4.* By Lemma B.3, we have

$$\nabla_u \frac{\partial L}{\partial p_t} = \nabla_u \mathbb{E}_S \big[ \nabla_{y'} \phi \big( f(p_t, \mathbf{x}), y \big) \alpha u h(\boldsymbol{\theta}, \mathbf{x}) \big] = -\widehat{g}_1(t, \boldsymbol{\theta}, u),$$

$$\nabla_{\boldsymbol{\theta}} \frac{\partial L}{\partial p_t} = \nabla_{\boldsymbol{\theta}} \mathbb{E}_S \big[ \nabla_{y'} \phi \big( f(p_t, \mathbf{x}), y \big) \alpha u h(\boldsymbol{\theta}, \mathbf{x}) \big] = -\widehat{g}_2(t, \boldsymbol{\theta}, u),$$

$$\nabla_u \frac{\partial D_{kL}(p_t||p_0)}{\partial p_t} = \nabla_u (\log(p_t/p_0) + 1) = u + \nabla_u \log(p_t),$$

$$\nabla_{\boldsymbol{\theta}} \frac{\partial D_{kL}(p_t||p_0)}{\partial p_t} = \nabla_{\boldsymbol{\theta}} (\log(p_t/p_0) + 1) = \boldsymbol{\theta} + \nabla_{\boldsymbol{\theta}} \log(p_t).$$

This proves the first four identities. For the last one, by the definition

$$\nabla \frac{\partial Q(p_t)}{\partial p_t} = \nabla \frac{\partial L(p_t)}{\partial p_t} + \lambda \nabla \frac{\partial D_{\mathrm{KL}}(p_t||p_0)}{\partial p_t},$$

we have

$$\nabla \cdot \left[ p_t(\boldsymbol{\theta}, u) \nabla \frac{\partial Q(p_t)}{\partial p_t} \right] = \nabla \cdot \left[ p_t(\boldsymbol{\theta}, u) \nabla \frac{\partial L}{\partial p_t} \right] + \lambda \nabla \cdot \left[ p_t(\boldsymbol{\theta}, u) \nabla \frac{\partial D_{\mathrm{KL}}(p_t||p_0)}{\partial p_t} \right]$$

$$= -\nabla_u \cdot [p_t(\boldsymbol{\theta}, u) \widehat{g}_1] - \nabla_{\boldsymbol{\theta}} \cdot [p_t(\boldsymbol{\theta}, u) \widehat{g}_2] + \lambda \nabla_u \cdot [p_t(\boldsymbol{\theta}, u) u]$$

$$\quad + \lambda \nabla_{\boldsymbol{\theta}} \cdot [p_t(\boldsymbol{\theta}, u) \boldsymbol{\theta}] + \lambda \nabla \cdot [p_t \nabla \log(p_t)]$$

$$= -\nabla_u \cdot [p_t(\boldsymbol{\theta}, u) g_1(t, \boldsymbol{\theta}, u)] - \nabla_{\boldsymbol{\theta}} \cdot [p_t(\boldsymbol{\theta}, u) g_2(t, \boldsymbol{\theta}, u)] + \lambda \Delta [p_t(\boldsymbol{\theta}, u)]$$

$$= \frac{dp_t}{dt},$$

where the third equation is by the definition $g_1(t, \boldsymbol{\theta}, u) = \widehat{g}_1(t, \boldsymbol{\theta}, u) - \lambda u$, $g_2(t, \boldsymbol{\theta}, u) = \widehat{g}_2(t, \boldsymbol{\theta}, u) - \lambda \boldsymbol{\theta}$ and $p_t \nabla \log(p_t) = \nabla p_t$. $\qquad \square$

### C.5 Proof of Lemma B.5

Here we give the proof of Lemma B.5.

*Proof of Lemma B.5.* The proof is based on the smoothness properties of $h(\boldsymbol{\theta}, \mathbf{x})$ given in Lemma A.1. We have

$$\big| \mathbb{E}_p \big[ \big( u h(\boldsymbol{\theta}, \mathbf{x}) + u \nabla h(\boldsymbol{\theta}, \mathbf{x}) \cdot \boldsymbol{\theta} - u \Delta h(\boldsymbol{\theta}, \mathbf{x}) \big) \big] \big|$$

$$\leq \mathbb{E}_p \big[ |u| G + G |u| \|\boldsymbol{\theta}\|_2 + G |u| \big]$$

$$= G \mathbb{E}_p [|u| \|\boldsymbol{\theta}\|_2] + 2 G \mathbb{E}_p [|u|]$$

$$\leq G \mathbb{E}_p \left[ \frac{u^2 + \|\boldsymbol{\theta}\|_2^2}{2} \right] + 2 G \sqrt{\mathbb{E}_p [u^2]},$$

where the first inequality is by $|h(\boldsymbol{\theta}, \mathbf{x})| \le G$, $\|\nabla_{\boldsymbol{\theta}} h(\boldsymbol{\theta}, \mathbf{x})\|_2 \le G$ and $|\Delta h(\boldsymbol{\theta}, \mathbf{x})| \le G$ in Lemma A.1, the second inequality is by Young's inequality and Cauchy-Schwartz inequality. Now by $\mathcal{W}(p, p_0) \le \sqrt{d+1}$ and Lemma B.1, we have

$$
\left| \mathbb{E}_p \Big[ \big( uh(\boldsymbol{\theta}, \mathbf{x}) + u \nabla h(\boldsymbol{\theta}, \mathbf{x}) \cdot \boldsymbol{\theta} - u \Delta h(\boldsymbol{\theta}, \mathbf{x}) \big) \Big] \right|
$$
$$
\le 2G(d+1) + 4G\sqrt{d+1}
$$
$$
= A_1.
$$

This completes proof. $\qquad\square$

## C.6 Proof of Lemma B.6

*Proof of Lemma B.6.* By Lemma B.3, we have

$$
\frac{\partial D_{\mathrm{KL}}(p_t \| p_0)}{\partial t} = \int_{\mathbb{R}^{d+1}} \frac{\partial D_{\mathrm{KL}}(p_t \| p_0)}{\partial p_t} \frac{dp_t}{dt} d\boldsymbol{\theta} du
$$
$$
= \int_{\mathbb{R}^{d+1}} \frac{\partial D_{\mathrm{KL}}(p_t \| p_0)}{\partial p_t} \nabla \cdot \left[ p_t(\boldsymbol{\theta}, u) \nabla \frac{\partial Q(p_t)}{\partial p_t} \right] d\boldsymbol{\theta} du
$$
$$
= -\int_{\mathbb{R}^{d+1}} p_t(\boldsymbol{\theta}, u) \left[ \nabla \frac{\partial D_{\mathrm{KL}}(p_t \| p_0)}{\partial p_t} \right] \cdot \left[ \nabla \frac{\partial Q(p_t)}{\partial p_t} \right] d\boldsymbol{\theta} du
$$
$$
= -\lambda \int_{\mathbb{R}^{d+1}} p_t(\boldsymbol{\theta}, u) \left[ \nabla \frac{\partial D_{\mathrm{KL}}(p_t \| p_0)}{\partial p_t} \right] \cdot \left[ \nabla \frac{\partial D_{\mathrm{KL}}(p_t \| p_0)}{\partial p_t} \right] d\boldsymbol{\theta} du
$$
$$
- \int_{\mathbb{R}^{d+1}} p_t(\boldsymbol{\theta}, u) \left[ \nabla \frac{\partial D_{\mathrm{KL}}(p_t \| p_0)}{\partial p_t} \right] \cdot \left[ \nabla \frac{\partial L(p_t)}{\partial p_t} \right] d\boldsymbol{\theta} du, \qquad \text{(C.1)}
$$

where the second and last equations are by Lemma B.4, the third inequality is by applying integration by parts multiple times. We further calculate by Lemma B.4,

$$
\int_{\mathbb{R}^{d+1}} p_t(\boldsymbol{\theta}, u) \left[ \nabla \frac{\partial D_{\mathrm{KL}}(p_t \| p_0)}{\partial p_t} \right] \cdot \left[ \nabla \frac{\partial D_{\mathrm{KL}}(p_t \| p_0)}{\partial p_t} \right] d\boldsymbol{\theta} du
$$
$$
= \int_{\mathbb{R}^{d+1}} p_t(\boldsymbol{\theta}, u) \| \boldsymbol{\theta} + \nabla_{\boldsymbol{\theta}} \log(p_t) \|^2 + \int_{\mathbb{R}^{d+1}} p_t(\boldsymbol{\theta}, u) | u + \nabla_u \log(p_t) |_2^2. \qquad \text{(C.2)}
$$

Moreover, for the second term on the right-hand side of (C.1) we have

$$
\int_{\mathbb{R}^{d+1}} p_t(\boldsymbol{\theta}, u) \left[ \nabla \frac{\partial D_{\mathrm{KL}}(p_t \| p_0)}{\partial p_t} \right] \cdot \left[ \nabla \frac{\partial L(p_t)}{\partial p_t} \right] d\boldsymbol{\theta} du
$$
$$
= \int_{\mathbb{R}^{d+1}} p_t(\boldsymbol{\theta}, u) [-\widehat{g}_1(t, \boldsymbol{\theta}, u)] \cdot [u + \nabla_u \log(p_t)] d\boldsymbol{\theta} du
$$
$$
+ \int_{\mathbb{R}^{d+1}} p_t(\boldsymbol{\theta}, u) [-\widehat{g}_2(t, \boldsymbol{\theta}, u)] \cdot [\boldsymbol{\theta} + \nabla_{\boldsymbol{\theta}} \log(p_t)] d\boldsymbol{\theta} du
$$
$$
= -\int_{\mathbb{R}^{d+1}} p_t(\boldsymbol{\theta}, u) [\widehat{g}_1(t, \boldsymbol{\theta}, u) \cdot u + \widehat{g}_2(t, \boldsymbol{\theta}, u) \boldsymbol{\theta}] d\boldsymbol{\theta} du
$$
$$
- \int_{\mathbb{R}^{d+1}} [\widehat{g}_1(t, \boldsymbol{\theta}, u) \cdot \nabla_u p_t(t, \boldsymbol{\theta}, u) + \widehat{g}_2(t, \boldsymbol{\theta}, u) \cdot \nabla_{\boldsymbol{\theta}} p_t(t, \boldsymbol{\theta}, u)] d\boldsymbol{\theta} du
$$
$$
= -\int_{\mathbb{R}^{d+1}} p_t(\boldsymbol{\theta}, u) [\widehat{g}_1(t, \boldsymbol{\theta}, u) \cdot u + \widehat{g}_2(t, \boldsymbol{\theta}, u) \boldsymbol{\theta}] d\boldsymbol{\theta} du
$$
$$
+ \int_{\mathbb{R}^{d+1}} p_t(t, \boldsymbol{\theta}, u) [\nabla_u \cdot \widehat{g}_1(t, \boldsymbol{\theta}, u) + \nabla_{\boldsymbol{\theta}} \cdot \widehat{g}_2(t, \boldsymbol{\theta}, u)] d\boldsymbol{\theta} du, \qquad \text{(C.3)}
$$

where the second equation is by $p_t \nabla \log(p_t) = \nabla p_t$ and the third equation is by applying integration by parts. Then plugging (C.2) and (C.3) into (C.1), we get

$$
\frac{\partial D_{\mathrm{KL}}(p_t \| p_0)}{\partial t} = -\lambda \int_{\mathbb{R}^{d+1}} p_t(\boldsymbol{\theta}, u) \| \boldsymbol{\theta} + \nabla_{\boldsymbol{\theta}} \log(p_t) \|_2^2 - \lambda \int_{\mathbb{R}^{d+1}} p_t(\boldsymbol{\theta}, u) | u + \nabla_u \log(p_t) |^2
$$
$$
+ \int_{\mathbb{R}^{d+1}} p_t(\boldsymbol{\theta}, u) [\widehat{g}_1 \cdot u + \widehat{g}_2 \cdot \boldsymbol{\theta} - \nabla_u \cdot \widehat{g}_1 - \nabla_{\boldsymbol{\theta}} \cdot \widehat{g}_2] d\boldsymbol{\theta} du.
$$

This completes the proof. $\qquad\square$

## C.7 Proof of Lemma B.7

*Proof of Lemma B.7.* We remind the readers the definitions of $\widehat{g}_1$ and $\widehat{g}_2$ in (A.1) and (A.1). We have

$$\int_{\mathbb{R}^{d+1}} p_t(\boldsymbol{\theta}, u)[\widehat{g}_1 \cdot u + \widehat{g}_2 \cdot \boldsymbol{\theta} - \nabla_u \cdot \widehat{g}_1 - \nabla_{\boldsymbol{\theta}} \cdot \widehat{g}_2] d\boldsymbol{\theta} du$$

$$= 2\alpha \mathbb{E}_S \left[ (f(p_t, \mathbf{x}) - y) \int_{\mathbb{R}^{d+1}} \big(uh(\boldsymbol{\theta}, \mathbf{x}) + u\nabla_{\boldsymbol{\theta}} h(\boldsymbol{\theta}, \mathbf{x}) \cdot \boldsymbol{\theta} - u\Delta h(\boldsymbol{\theta}, \mathbf{x})\big) p_t(\boldsymbol{\theta}, u) d\boldsymbol{\theta} du \right].$$

Denote $I(\boldsymbol{\theta}, u, \mathbf{x}) = uh(\boldsymbol{\theta}, \mathbf{x}) + u\nabla_{\boldsymbol{\theta}} h(\boldsymbol{\theta}, \mathbf{x}) \cdot \boldsymbol{\theta} - u\Delta h(\boldsymbol{\theta}, \mathbf{x})$, then we have

$$|\nabla_u I(\boldsymbol{\theta}, u, \mathbf{x})| = |h(\boldsymbol{\theta}, \mathbf{x}) + \nabla_{\boldsymbol{\theta}} h(\boldsymbol{\theta}, \mathbf{x}) \cdot \boldsymbol{\theta} - \Delta h(\boldsymbol{\theta}, \mathbf{x})| \leq G\|\boldsymbol{\theta}\|_2 + 2G, \qquad (C.4)$$

where the inequality holds by Lemma A.1. Similarly, we have

$$\|\nabla_{\boldsymbol{\theta}} I(\boldsymbol{\theta}, u, \mathbf{x})\|_2 = \|u\nabla_{\boldsymbol{\theta}} h(\boldsymbol{\theta}, \mathbf{x}) + u\nabla_{\boldsymbol{\theta}}\big(\nabla_{\boldsymbol{\theta}} h(\boldsymbol{\theta}, \mathbf{x}) \cdot \boldsymbol{\theta}\big) - u\nabla_{\boldsymbol{\theta}} \Delta_{\boldsymbol{\theta}} h(\boldsymbol{\theta}, \mathbf{x}))\|_2$$

$$\leq 3G|u|. \qquad (C.5)$$

Therefore, combining the bounds in (C.4) and (C.5) yields

$$\sqrt{\nabla_u I(\boldsymbol{\theta}, u, \mathbf{x})^2 + \|\nabla_{\boldsymbol{\theta}} I(\boldsymbol{\theta}, u, \mathbf{x})\|_2^2} \leq 4G\sqrt{u^2 + \|\boldsymbol{\theta}\|_2^2} + 2G.$$

By Lemma B.2, we have that

$$\mathbb{E}_{p_t}[I(\boldsymbol{\theta}_t, u_t, \mathbf{x})] - \mathbb{E}_{p_0}[I(\boldsymbol{\theta}_0, u_0, \mathbf{x})] \leq \left[8G\sqrt{d+1} + 2G\right]\mathcal{W}(p_0, p_t)$$

$$\leq A_2\sqrt{D_{\mathrm{KL}}(p_t\|p_0)},$$

where the last inequality is by Lemma 5.4 and $A_2 = 16G\sqrt{d+1} + 4G$. By $\mathbb{E}_{p_0}[I(\boldsymbol{\theta}_0, u_0, \mathbf{x})] = \mathbb{E}_{p_0}[u_0]\mathbb{E}_{p_0}[h(\boldsymbol{\theta}_0, \mathbf{x}) + \nabla_{\boldsymbol{\theta}} h(\boldsymbol{\theta}_0, \mathbf{x}) \cdot \boldsymbol{\theta}_0 - \Delta_{\boldsymbol{\theta}} h(\boldsymbol{\theta}_0, \mathbf{x})] = 0$, we further have

$$\mathbb{E}_{p_t}[I(\boldsymbol{\theta}_t, u_t, \mathbf{x})] \leq A_2\sqrt{D_{\mathrm{KL}}(p_t\|p_0)}. \qquad (C.6)$$

Then we have

$$\int_{\mathbb{R}^{d+1}} p_t(\boldsymbol{\theta}, u)[\widehat{g}_1 \cdot u + \widehat{g}_2 \cdot \boldsymbol{\theta} - \nabla \cdot \widehat{g}_1 - \nabla \cdot \widehat{g}_2] d\boldsymbol{\theta} du$$

$$= 2\alpha \mathbb{E}_S \left[ (f(p_t, \mathbf{x}) - y)\mathbb{E}_{p_t}[I(\boldsymbol{\theta}_t, u_t, \mathbf{x})] \right]$$

$$\leq 2\alpha A_2 \sqrt{D_{\mathrm{KL}}(p_t\|p_0)}\sqrt{L(p_t)},$$

where the last inequality is by (C.6) and Cauchy-Schwarz inequality. This completes the proof. $\qquad \square$

## Footnotes

[4]The derivation we present here works as long as $L(p_t) \neq 0$. A more thorough but complicated analysis can deal with the case when $L(p_t) = 0$ for some $t$. However for simplicity we omit the more complicated proof, since loss equaling to zero is a trivial case for a learning problem.