[Reviews · NeurIPS 2020]

Review 1

Summary and Contributions: This paper studies optimization and generalization of a two-layer, infinite width neural network with a three-times differentiable activation function, weights decaying, and trained by noisy gradient descent. The considered scaling factor covers the mean field theory and neural tangent kernel regime.

Strengths: - extends result of (Mei et al. 2019) with gradient noises and weight decay regularizer - provides generalization properties (estimation error bounds) of such two-layer neural network

Weaknesses: After checked the rebuttal, the authors address my concerns on alpha = poly(n,d,lambda). I increase my score to 6. ============================================================== The quality of this paper is ok, but I'm doubting on the used assumptions/conditions. - One key issue is regarding to the minimal eigenvalue of the Gram matrix in NTK that largely effects the trainable and generalization properties. As indicated by Thm 2 in [S1], the minimal eigenvalue of NTK converges at O(n^-1/2) rate, so in this case, \lambda_0 \propto n^{-3/4}. Or, it scales at a constant order O(1), which implies \lambda_0 \propto n^{-1/2}. This deserves a detailed discussion since it indeed effects the presented theorems to avoid some unattainable cases. [S1] On Learning Over-parameterized Neural Networks: A Functional Approximation Perspective. NeurIPS, 2019. For example, 1) Thm 4.4: if \lambda_0 := n^{-3/4}, the condition (4.1) in Thm 4.4 acutally is alpha > \lambda^{1/2} n^{2/3}, which appears difficult to cover the mean filed case with alpha = 1. This is because, in learning theory, the regularization parameter is taken by \lamda := n^{-a} with 0 <a \leq 1. This issue also exists in Thm 4.5 with alpha > sqrt(n \lambda). 2) Thm 4.4 (also Lemma 5.3) D_{KL}(p_t || p_0) < O(alpha^{-2} \lambda_0^{-4}) appears to increase with n. This divergence looks strange to me, and how to explain this? 3) The above result on D_{KL}(p_t || p_0) < O(alpha^{-2} \lambda_0^{-4}) makes me scrutinize the assumption D(p_true || p_0) < +\inf in Thm 4.5. I'm not sure that this assumption is attainable. Overall, it's important to discuss whether the used conditions and assumptions in the presented results are attaianble/mild or not. I'm ready to read the author's argument about them.

Correctness: The presented results appear technically sound to me, although I did not check them very carefully. Nevertheless, I'm doubting about the used conditions/assumptions.

Clarity: The paper is well written and easy to follow.

Relation to Prior Work: This paper clarified the difference between their work and previous work.

Reproducibility: Yes

Additional Feedback:


Review 2

Summary and Contributions: Prior works have established convergence of gradient descent on sufficiently wide two-layer neural networks. This paper aims to extend the same result to cases with noisy gradient and weight decay regularization.

Strengths: Convergence of gradient descent with Gaussian noise and regularization is proved, under certain assumptions. Moreover, the convergence rate is shown to be linear.

Weaknesses: The authors claim without justification that the standard tools used in NTK regimes CANNOT handle noisy gradient and regularizers (Line 28 and 168). It is not clear that whether the prior works just did not analyze these two particular scenarios due to limited space or similar, or these two scenarios cannot be handled in principle. If it is the first case, the results in this paper would be not quite significant, and can be considered as a natural extension of previous works. So, I suggest the author to spend a section or so to provide detailed analysis on how and why noisy gradient and regularizers can not be handled by standard NTK analysis. Note that the model f contains a scaling factor alpha, see Eq.(3.1) and (3.2). According to the definition of tangent kernel, the scaling factor alpha should appear in the expression (line 127), but it is missing. This is important, because a small scaling factor also scale the kernel matrix, and fails the assumption 4.3, which is the base of the main theorem. In Line 130, it is not quite exact to claim “parameters stay close to initialization” without mentioning in what norm. When consider the NTK regime (as in Jacot et al), i.e., the sufficiently wide neural networks, it is important to distinguish the infinity-norm and Euclidean norm, when the number of parameters is large. As pointed out by Liu et al (reference [1] below), parameters do NOT stay close to initialization in the sense of Euclidean norm (see Remark 6.2 therein), although the infinity-norm is small. This is because a large number of small componentwise changes adds up to a non-small total change. The gradient noise and regularizer are controlled by a common coefficient \lambda. In principle, gradient noise and regularizer are totally independent and should have different coefficients. My question is: do the results still hold, if the gradient noise and regularizer are controlled by different coefficient? Reference: [1] Liu, C., Zhu, L., and Belkin, M., Toward a theory of optimization for over-parameterized systems of non-linear equations: the lessons of deep learning. Arxiv: 2003.00307, (2020).

Correctness: I have a little doubt about the correctness of the expression in line 127. As I mentioned in the weakness section, a scaling factor may be missing.

Clarity: As for Theorem 4.4, I suggest the authors to provide some intuitions on the following: >>how does the scaling factor alpha affect the results? Why does scaling factor matter? From Eq. (4.1), it seems that for smaller alpha the theorem does not hold and convergence is not guaranteed for the noisy gradient and regularization scenarios. >>A discussion of the order of scaling factor alpha is preferred.

Relation to Prior Work: Related works are clearly discussed in a separate section.

Reproducibility: Yes

Additional Feedback:


Review 3

Summary and Contributions: This paper addresses one of the limitations of the NTK analysis in handling regularization terms (e.g., ell_2). Instead of studying gradient flow in the parameter space, the authors base upon mean-field analysis and transform the parameter learning into a distribution learning in the space of probability measures. An important ingredient is additive Gaussian noises to gradient updates. This allows the authors to achieve a linear convergence of the squared loss. They also establish a generalization bound.

Strengths: The results are a non-trivial extension of the standard NTK analysis and new. The proof of convergence (Theorem 4.5) is very similar in spirit with the gradient flow on the loss, where instead of controlling the movement of the weights, this paper focuses on the Wasserstein distance between p_t and p_0. One interesting thing is the dynamics of the regularization term KL(p||p0) is coupled with the dynamics of the loss with respect to the measure p_t, which is not possible in standard NTK.

Weaknesses: One limitation with this mean-field analysis is that there is no finite bound on the number of neurons. Moreover, it seems to be difficult to extend this analysis for non-smooth activation, for example RELU.

Correctness: I didn’t check all the proof in Appendix, but the statements appear sensible and correct.

Clarity: Yes

Relation to Prior Work: Yes

Reproducibility: Yes

Additional Feedback:


Review 4

Summary and Contributions: The paper proposed a generalized analysis for neural tangent kernel so that it can address the empirical practices of regularization and gradient noises during training process.

Strengths: The paper proved the generalization performance of the noisy gradient descent algorithm with regularization. It tries to address the concern that weight distribution is no longer close to the initialization, when regularization is used during optimization.

Weaknesses: One of important point of NTK was over-parameterization. A lot of research work recently is showing regularization is not critical for over-parameterized models. Even though the authors talk a little bit about the necessity of weight decay in Sec 3, I'm still not quite convinced about the motivation of paper about using regularization in NTK. It might weaken the possible implications of the theoretical results in the paper.

Correctness: The theoretical results should be correct. Assumption 4.2 seems a little bit strong, as some popular activation functions aren't really 3-times differentiable, e.g. Relu-type activations. It limited the applicability of the theoretical results. The assumption in 4.5 of existence of p_true also seems non-trivial. But it might make sense in practices as the training error usually converges to 0. Maybe authors could discuss a little bit more about this assumption as well.

Clarity: The paper made the main point clearly overall. Some of the argument in Section 3 is a little bit too skechy and the authors could provide more details. E.g. * The paper could make the derivation of Eqn (3.4) more explicitly. * The connection from Eqn (3.4) and (3.5) is also non-trivial. They did provide some references, but it'll also be good to give brief recap to make the paper more readable.

Relation to Prior Work: The paper discussed a good collection of related works about neural tangent kernel and related theoretical analysis. There are a couple of points I also wanted the authors to discuss: * The noisy gradient seems a rough approximation of the stochastic gradient descent algorithm commonly used in practices. There's some existing work analyzing the generalization bound for SGD directly. Could the authors discuss a little bit more on this? * The author claims that weight decay regularization is still meaning in the neural tangent kernel regime. But in the literature, there're some discussion that regularization might not be necessary for over-parameterized model. I'd like the authors to elaborate more on this point, as it's critical to the main point for this paper.

Reproducibility: Yes

Additional Feedback: ********edit************** I updated my score since the paper is well-written and the point is made clearly.

[Author Response · NeurIPS 2020]

We thank all the reviewers for their detailed and helpful comments.

**Response to Reviewers 1&4:** Is the assumption $D_{\mathrm{KL}}(p_{\mathrm{true}}||p_0) < +\infty$ in Thm 4.5 attainable?

**R:** We would like to clarify that the definition of $p_{\mathrm{true}}$ is to satisfy $y = \int uh(\boldsymbol{\theta}, \mathbf{x})p_{\mathrm{true}}(\boldsymbol{\theta}, u)d\boldsymbol{\theta}du$, which is ir-
relevant to $\alpha$. In comparison, $p_t$ is the parameter distribution for the neural network defined in (3.1) with the
scaling factor $\alpha$. So the KL-divergence bound on $p_t$ in Theorem 4.4 does not contradict with the existence of
$p_{\mathrm{true}}$. This assumption on $p_{\mathrm{true}}$ essentially assumes that the target function is in the very big function class
$\mathcal{F} = \{f(\mathbf{x}) = \int uh(\boldsymbol{\theta}, \mathbf{x})p_{\mathrm{true}}(\boldsymbol{\theta}, u)d\boldsymbol{\theta}du, D_{\chi^2}(p_{\mathrm{true}}||p_0) < +\infty\}$, and therefore it is attainable. Note that this
type of assumption on the target function is inevitable: Without any target function assumptions, the random label case
is not excluded, and for random labels small test error is impossible. We will add more discussion in the camera ready.

**Response to Reviewer 1:**

**Q1:** The minimal eigenvalue of the NTK Gram matrix affects trainable and generalization properties

**R1:** Thank you for pointing out the related work [S1]. We will cite this paper and add more discussion on the rate of
the smallest eigenvalue of NTK Gram matrix in the camera ready. $\lambda_0$ indeed depends on $n$, which is consistent with
existing NTK literature. However, our theorem assumptions are all still attainable in this setting.

**Q2(a):** (4.1) in Thm 4.4 cannot cover the mean filed case $\alpha = 1$. **(b):** The KL bound in Thm 4.4 increases with $n$.

**R2:** We believe both questions are caused by a misunderstanding on the scaling factor $\alpha$. We would like to clarify that
$\alpha$ is not $O(1)$. Instead, in all of our main results (Theorems 4.4 and 4.5), $\alpha = \mathrm{poly}(n)$. In other words, Theorem 4.4 is
not supposed to cover the mean field case $\alpha = 1$ at all. Also, because of $\alpha = \mathrm{poly}(n)$, the KL bound in Theorem 4.4
will not increase with $n$. A simple way to parse and understand our results is to make an analogy between $\alpha$ and the
square root of network width $\sqrt{m}$ in the standard NTK literature.

**Response to Reviewer 2:**

**Q1:** Explain how and why noisy gradient and regularizers can not be handled by standard NTK analysis

**R1:** Thank you for your suggestion. We will add a section to explain this claim in detail. Here we provide a short
explanation. In noisy gradient descent, the weight decay regularizer pushes the weights towards zero, and gradient
noises further push the weights towards a random direction. Therefore, they jointly make each weight fairly far away
from initialization. However, the joint effect of weight decay and gradient noise does not push the distribution far away
from initialization, as they together give a KL-divergence regularization in the energy functional.

**Q2:** The scaling factor should appear in the definition of tangent kernel

**R2:** Our definition of the NTK is correct and consistent with existing results. Our definition matches the definition in
equation (16) in [25], where a similar large scaling factor is also considered.

**Q3:** Should specify in what sense does "parameters stay close to initialization".

**R3:** Here by "parameters stay close to initialization" we mean the "node-wise" $\ell_2$-norm distance. Thanks for pointing
out the related work. We will comment on it in the camera ready.

**Q4:** Do the results still hold, if the gradient noise and regularizer are controlled by different coefficient?

**R4:** When the gradient noise and regularizer scales are different, the corresponding regularizer on distribution is no
longer on the KL-divergence towards initialization distribution $p_0$, but is towards some different Gaussian distribution $\widetilde{p}$.
Therefore, this setting is likely different from the NTK regime, and our linear convergence result may no longer hold.
Nevertheless, our generalization results can easily cover this setting by assuming $D_{\mathrm{KL}}(p_{\mathrm{true}}||\widetilde{p}) < +\infty$.

**Q5:** How does the scaling factor alpha affect the results? Why does scaling factor matter? From Eq. (4.1), it seems that
for smaller alpha the theorem does not hold. A discussion of the order of scaling factor alpha is preferred

**R5:** As we discussed around line 94, $\alpha$ corresponds to the square root of network width in the standard NTK regime,
and therefore condition (4.1) is the counterpart of the network width requirement $m \geqslant \mathrm{poly}(n)$ in standard NTK-type
optimization results [2,14,15,35]. Therefore, requiring a large $\alpha$ is natural to ensure that the training of the network is in
the NTK regime (or lazy training regime), and the setting with smaller $\alpha$ is not the focus of this paper. We will clarify it
and provide the specific order of $\alpha$ in the camera ready.

**Response to Reviewer 3:**

**Q1:** There is no finite bound on the number of neurons.

**R1:** By using similar techniques as in [25,26], we are able to study how the training of a finitely wide network can be
approximated by the PDE (3.4) in a bounded time interval $[0, T]$. We will add more discussion in the camera ready.

**Response to Reviewer 4:**

**Q1:** motivation of regularization, regularization might not be necessary for over-parameterized model

**R1:** Weight decay regularization is a widely used regularization in deep learning practice, and therefore we believe it
is important to establish theoretical guarantees that cover weight decay. It is true that generalization bounds can be
developed even without the use of regularizers. This is mainly due to the study of the implicit regularization induced by
training algorithms. However, the study of explicit regularization is still an important problem, as explicit regularization
can still affect generalization in a different way compared with implicit regularization. See e.g., [33].

**Q2:** Discuss on the relation to generalization bounds for SGD

**R2:** Our generalization bound is in the probability measure space. In comparison, existing generalization bounds for
SGD are in parameter space, which is not applicable in our setting.

[Meta-Review · NeurIPS 2020]

This paper extends neural tangent kernel results to a two-layer, infinite width neural network with a three-times differentiable activation function, weight decay regularization, and noisy gradient descent training, showing a linear convergence rate. The paper received mixed reviews (marginally above, marginally below, accept, reject). On the positive side, R3 think the results are a new nontrivial extension of the NTK results, and R1 think the paper is novel, well written, etc. R1 had some technical issues, but was satisfied by the rebuttal. On the other hand, R2 raised some technical issues regarding the effect of the scaling in the kernel, which I think are well addressed by the rebuttal. R4's main critique is that he/she is not convinced about the significance of using L2 regularization, since algorithms have implicit regularization. I am satisfied by the rebuttal, which essentially argues that both implicit and explicit regularization have value and should be studied.